# Beyond Geometry: Functionally Grounded Molecule Generation for Structure-Based Drug Design

## Abstract

Structure-based drug design aims to generate 3D ligands that bind stably to specific protein pockets. While recent generative models have improved by incorporating the geometry of protein pockets, they still overlook the biochemical functional interactions between proteins and ligands. Crucial interactions include hydrogen bonds, hydrophobic interactions, and $\pi$-$\pi$ stacking, which are essential for binding affinity and structural stability. This oversight leads to strained, high-energy ligands that may geometrically fit but functionally misalign with the binding site. To bridge the gap, we introduce a **F**unctionally **G**rounded **Mol**ecule Generation Network (**FGMol**) that operates in a unified structure-function alignment framework, enabling molecular generation to align with protein-ligand interactions, extending beyond mere geometric fitting. Our design of FGMol introduces: (1) Interaction-Aware Embedding, which annotates protein atoms with explicit interaction types and feed them into SE(3)-equivariant neural networks; (2) Interaction-Informed Motif Alignment, which leverages differentiable clustering and Sinkhorn matching to align protein-ligand functional motifs; and (3) Interaction-Guided Generation with Bayesian Flow Network, which independently models ligand coordinates and atom types via Bayesian updates in continuous space, conditioned on protein-guided cross-attention. Experiments on the CrossDocked2020 benchmark demonstrate that FGMol surpasses prior state-of-the-art methods in binding affinity, and notably reduces strain energy by over 20%, while maintaining high synthetic accessibility—highlighting its advantage in interaction-aware ligand generation.

## 1 Introduction

Structure-based drug design (SBDD) (Anderson, 2003; Du et al., 2024; Kuntz, 1992) guides the rational design of 3D ligands that bind to protein targets with high binding affinity and specificity (Batool et al., 2019; Gebauer et al., 2019; Lin et al., 2025). Early autoregressive methods in SBDD generate ligands either atom-by-atom (Luo et al., 2021; Peng et al., 2022) or fragment-by-fragment (Fu et al., 2025; Zhang et al., 2023c), but their lack of global structural context leads to serious error accumulation (Guan et al., 2023a). To address this, diffusion-based models (Ho et al., 2020) independently model atom types and positions in a non-autoregressive manner (Guan et al., 2023a; Schneuing et al., 2024), but they often suffer from slow sampling and complex hybrid denoising schemes (Qu et al., 2024). Recently, flow-based methods shift away from conventional denoising paradigms by modeling ligand generation as a direct transport process, significantly improving both efficiency and structural fidelity (Qu et al., 2024; Zhang et al., 2024; Zhou et al., 2025).

Despite these advances, most existing methods (Guan et al., 2023a;b; Luo et al., 2021; Peng et al., 2022; Qu et al., 2024) remain focused on fitting the geometric shape of protein pockets while largely overlooking the underlying functional interactions, such as hydrogen bonds, hydrophobic interactions, and $\pi$-$\pi$ stacking (Figure 1) (Guedes et al., 2018; Kitchen et al., 2004; Lin et al., 2023). However, these interactions play a decisive role in determining whether a ligand binds effectively to a protein pocket. For instance, hydrogen bonds contribute directional specificity and binding energy; hydrophobic contacts stabilize ligands in nonpolar cavities; and $\pi-\pi$ interactions are crucial for anchoring aromatic moieties in planar regions of the pocket (Patil et al., 2010). Ignorance of those biochemical cues renders current models incapable of positioning (Guan et al., 2023a; Luo et al.,

2021). As a result, the solely geometry-based models generate strained or misaligned conformations that fail to capture the intended interaction patterns. The inaccuracies ultimately undermine binding affinity, structural stability, and synthetic accessibility (Shoichet et al., 2002; Choi et al., 2024).

To address these challenges, we argue that ligand generation should go beyond geometric fitting and be guided by the functional interactions at protein binding sites. This raises two key questions: 1) How can a model perceive the pocket's functional interactions, such as spotting where hydrogen bonding and hydrophobic interactions are preferred (Sako et al., 2024)? Most existing methods (Guan et al., 2023a; Qu et al., 2024) overlook these decisive binding interactions and focus solely on geometric shape. 2) How can a model ensure that atoms are not only placed correctly but also remain stable? While many methods can predict plausible locations, they often fail to ensure the physical plausibility and chemical stability of those placements, resulting in strained, high-energy conformations that ultimately undermine synthetic feasibility.

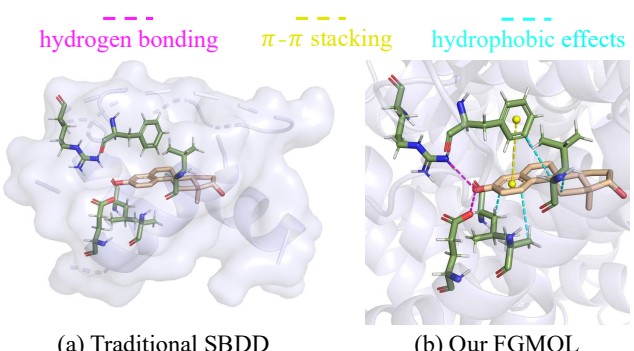

(a) Traditional SBDD    (b) Our FGMOL

Figure 1: Comparison between traditional SBDD and our FGMOL. (a) Traditional SBDD focuses on protein geometry and overlooks intrinsic functional interactions. (b) Our FGMOL explicitly models key functional interactions—hydrogen bonding, $\pi - \pi$ stacking, and hydrophobic effects—for structure-function-based molecular design. In both panels, the orange structure represents the ligand, while the gray and green regions correspond to the protein.

Herein, we introduce **FGMOL**, a unified framework that bridges protein structure, functional interactions, and molecular generation within a continuous parameter space. FGMOL consists of three key components: **(1) Interaction-Aware Embedding**: We annotate each protein atom with explicit interaction types (e.g., hydrogen bond donor/acceptor, hydrophobic), encoded as a multi-hot vector. These annotations are fused with 3D spatial features via an SE(3)-equivariant neural network (Fuchs et al., 2020), enabling the model to reason about the biochemical roles of the binding site. **(2) Interaction-Informed Motif Alignment**: to capture higher-order pharmacophoric-like motifs, the protein and ligand atoms are clustered and aligned via function-aware Sinkhorn normalization (Sinkhorn & Knopp, 1967). This enforces functional complementarity and stabilizes motif-level alignment. **(3) Interaction-Guided Generation with a Bayesian Flow Network (BFN)** (Graves et al., 2023): ligand generation is modeled as a trajectory in continuous space, guided by Bayesian posterior updates conditioned on structure-function alignment. A cross-attention mechanism injects functional knowledge at each generation step, promoting structures that are chemically stable, physically plausible, and synthetically feasible. Our main contributions are:

- We propose **FGMOL**, a unified framework that bridges protein structure, functional interactions, and ligand generation within a continuous parameter space, extending structure-based drug design beyond pure geometric fitting toward interaction-driven molecule generation.

- We introduce a structure-function alignment framework by annotating protein atoms with explicit functional interaction types and aligning protein and ligand motifs via differentiable pooling and Sinkhorn normalization, providing biochemical grounding for ligand generation.

- We develop a Bayesian Flow Network-based molecule generator that independently models atomic coordinates and types in continuous space, injecting functional priors via cross-attention, enabling the generation of biochemically plausible and structurally stable ligands.

- We demonstrate that FGMOL achieves state-of-the-art performance on the CrossDocked2020 benchmark, outperforming baselines across multiple criteria including binding affinity, strain energy, synthetic accessibility, and functional interaction recovery.

## 2 PRELIMINARY

**Notations and Problem Formulation**    The structure-based drug design (SBDD) task is defined as generating a ligand $\mathcal{M} = \left\{ \left( \mathbf{x}_m^{(i)}, \mathbf{v}_m^{(i)} \right) \right\}_{i=1}^{N_m}$ to fit a given protein pocket $\mathcal{P} = \left\{ \left( \mathbf{x}_p^{(i)}, \mathbf{v}_p^{(i)} \right) \right\}_{i=1}^{N_p}$, where $N_p$ and $N_m$ denote the number of atoms in the protein and ligand, respectively. Here, $\mathbf{x}^{(i)} \in \mathbb{R}^3$ denotes the 3D coordinates of an atom, and $\mathbf{v}^{(i)} \in \mathbb{R}^K$ represents its type. To simplify notation, the ligand molecule is represented as $\mathbf{m} = [\mathbf{x}_m, \mathbf{v}_m]$, where $\mathbf{x}_m \in \mathbb{R}^{N_m \times 3}$ and $\mathbf{v}_m \in \mathbb{R}^{N_m \times K}$. Similarly, the protein is represented as $\mathbf{p} = [\mathbf{x}_p, \mathbf{v}_p]$, where $\mathbf{x}_p \in \mathbb{R}^{N_p \times 3}$ and $\mathbf{v}_p \in \mathbb{R}^{N_p \times K}$. In addition, we incorporate functional interaction priors for the protein atoms, denoted as $\mathbf{a} \in \mathbb{R}^{N_p \times N_a}$, where $N_a$ is the number of interaction types. These priors provide biochemical cues that extend beyond geometric structure to guide ligand generation. Thus, the SBDD task in this work is to model $p(\mathbf{m} \mid \mathbf{p}, \mathbf{a})$, where generation is guided by both the protein geometry and functional interactions.

**Continuous Molecule Generation with Bayesian Flow Network**    Inspired by Song et al. (2024), the atomic coordinates and types are independently modeled in continuous parameter space. Building on this, the sender-receiver paradigm of the Bayesian Flow Network (BFN) (Graves et al., 2023) is adopted as the generation framework, guiding the training process and molecular sampling.

**Unified parameter $\boldsymbol{\theta} \stackrel{\text{def}}{=} [\boldsymbol{\theta}^x, \boldsymbol{\theta}^v]$.** The atomic coordinates and types are independently modeled using a continuous probabilistic formulation, denoted as $\boldsymbol{\theta} := [\boldsymbol{\theta}^x, \boldsymbol{\theta}^v]$, where $\boldsymbol{\theta}^x = \{\boldsymbol{\mu}, \rho\}$ and $\boldsymbol{\theta}^v \in \mathbb{R}^{N_m \times K}$. Specifically, continuous atom coordinates $\mathbf{x}$ are modeled as a multivariate Gaussian distribution $\mathcal{N}(\boldsymbol{\mu}, \rho^{-1}\mathbf{I})$, while discrete atom types $v$ are relaxed into categorical distributions parameterized by $\boldsymbol{\theta}^v$. The Bayesian updates for $\rho$, $\boldsymbol{\mu}$ and $\boldsymbol{\theta}^v$ are given by:

$$\rho_i = \rho_{i-1} + \alpha_i, \qquad \boldsymbol{\mu}_i = \frac{\boldsymbol{\mu}_{i-1}\rho_{i-1} + \mathbf{y}^x \alpha_i}{\rho_i}, \qquad \boldsymbol{\theta}_i^v = \frac{e^{\mathbf{y}^v} \boldsymbol{\theta}_{i-1}^v}{\sum_{k=1}^K e^{\mathbf{y}_k^v}(\boldsymbol{\theta}_{i-1}^v)_k}, \quad (1)$$

where $\mathbf{y}^x$ and $\mathbf{y}^v$ are Gaussian noise samples drawn from the sender distribution, and $\alpha_i$ denote noise factor at step $i$. For priors $\boldsymbol{\theta}_0$, we adopt standard Gaussian and uniform distribution respectively.

**Sender-Receiver Paradigm in BFN.** With the unified parameterization, BFN follows a sender-receiver message exchange paradigm (Qu et al., 2024). At each timestep $t_i$, the sender operates in the sample space by selecting a molecule $\mathbf{m}$ and adding Gaussian noise $\mathbf{y}_i$, drawn from the sender distribution $p_S(\mathbf{y}_i \mid \mathbf{m}; \alpha_i)$, where the $\alpha_i$ is the noise factor from the schedule $\beta(t_i)$. The resulting noisy latent is then passed to the receiver like the forward diffusion process. The receiver uses the previous parameters $\boldsymbol{\theta}_{i-1}$, the protein context $\mathbf{p}$, and the known noise factor $\mathbf{a}$ to sample $\hat{\mathbf{m}}$ from the output distribution $p_O$, and then predicts the latent noise $\hat{\mathbf{y}}_i$.

Specifically, the sender distribution is defined as:

$$p_S\left(\mathbf{y}^x \mid \mathbf{x}_m; \alpha\right) = \mathcal{N}\left(\mathbf{y}^x \mid \mathbf{x}_m, \alpha^{-1}\mathbf{I}\right), \quad p_S\left(\mathbf{y}^v \mid \mathbf{v}_m; \alpha'\right) = \mathcal{N}\left(\mathbf{y}^v \mid \alpha'\left(K\mathbf{e}_{\mathbf{v}_m} - \mathbf{1}\right), \alpha' K\mathbf{I}\right), \quad (2)$$

where $\mathbf{e}_{\mathbf{v}_m} = [\mathbf{e}_{\mathbf{v}^{(1)}}, \dots, \mathbf{e}_{\mathbf{v}^{(N_m)}}] \in \mathbb{R}^{N_m \times K}, \mathbf{e}_j \in \mathbb{R}^K$ is a one-hot vector.

And the receiver distribution is defined as:

$$p_R\left(\mathbf{y}^x \mid \boldsymbol{\theta}^x, \mathbf{p}; t\right) = \mathcal{N}\left(\mathbf{y}^x \mid \boldsymbol{\Phi}\left(\boldsymbol{\theta}^x, \mathbf{p}, t\right), \alpha^{-1}\mathbf{I}\right), \quad p_R\left(\mathbf{y}^v \mid \boldsymbol{\theta}^v, \mathbf{p}; t\right) = \left[p_R\left((\mathbf{y}^v)^{(d)} \mid \cdot\right)\right]_{d=1\dots N}, \quad (3)$$

where $p_R\left((\mathbf{y}^v)^{(d)} \mid \cdot\right) = \sum_k p_O^v(k \mid \cdot) p_S^v\left((\mathbf{y}^v)^{(d)} \mid k; \alpha\right)$. And the predicted structure $\hat{\mathbf{m}}$ is sampled from $p_O(\cdot \mid \boldsymbol{\theta}_{i-1}, \mathbf{p}, t_i)$, and $p_O$ is parameterized by a neural network $\boldsymbol{\Phi}(\boldsymbol{\theta}_{i-1}, \mathbf{p}, t_i)$.

**Training Objective.** To enable simulation-free optimization, we adopt the Bayesian flow distribution $p_F$ following Graves et al. (2023), which marginalizes the recursive update process into a closed-form:

$$p_F(\boldsymbol{\theta}_i \mid \mathbf{m}, \mathbf{p}; t_i) = \mathbb{E}_{\boldsymbol{\theta}_{1\dots i-1} \sim p_U} p_U(\boldsymbol{\theta}_i \mid \boldsymbol{\theta}_{i-1}, \mathbf{m}, \mathbf{p}; \alpha_i) = p_U(\boldsymbol{\theta}_i \mid \boldsymbol{\theta}_0, \mathbf{m}, \mathbf{p}; \beta(t_i)), \quad (4)$$

where $p_U$ denotes the Bayesian update distribution derived from the recursive update rule $h$. The training objective is to minimize the expected KL divergence between sender and receiver distributions over $n$ inference steps:

$$\mathcal{L}^n(\mathbf{m}, \mathbf{p}) = \mathbb{E}_{i \sim \mathcal{U}(1,n), \mathbf{y}_i \sim p_S, \boldsymbol{\theta}_{i-1} \sim p_F} D_{\text{KL}}\left(p_S \parallel p_R\right), \quad (5)$$

where $\mathcal{U}(1, n)$ denotes uniform timestep sampling. This KL loss encourages the receiver to approximate the reverse process, progressively denoising the sample to align with the posterior.

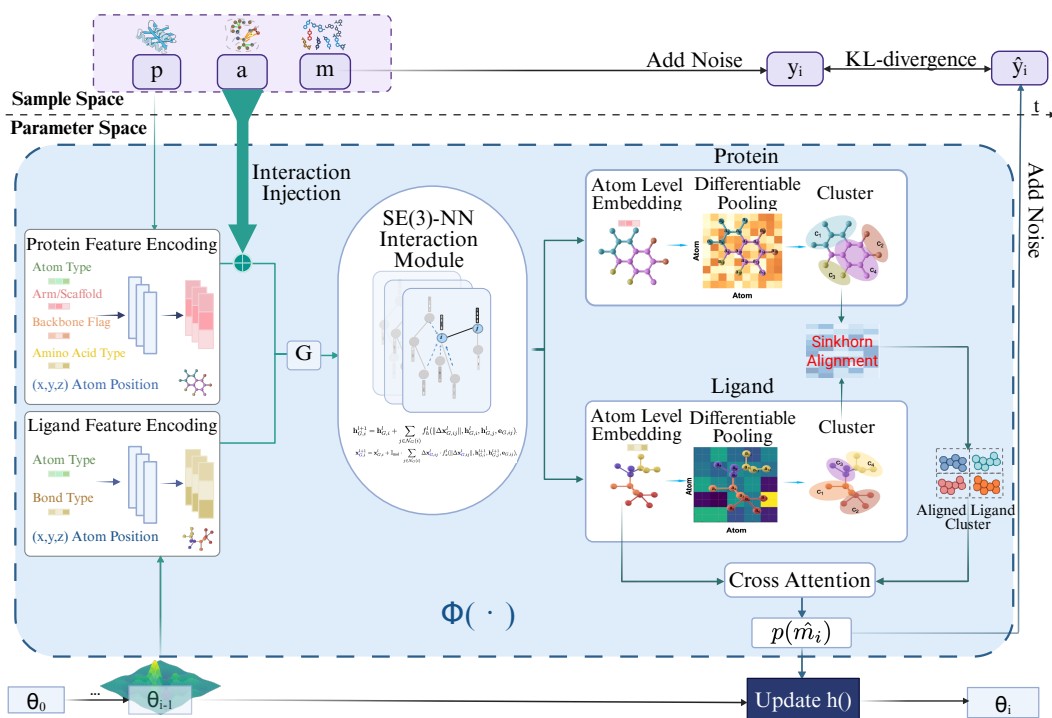

Figure 2: The overall architecture integrates a BFN (outer loop) with an aligned structure-function generative model (inner blue block) in a three-stage pipeline, indicated by arrows: light green for interaction-aware embedding, dark green for interaction-informed motif alignment, and black for interaction-guided generation with BFN. First, interaction types are embedded into protein atoms and fused with 3D features via an SE(3)-equivariant network. Second, protein and ligand motifs are aligned using Sinkhorn normalization to ensure functional complementarity. Third, ligands are generated in continuous space, guided by functional priors through cross-attention.

## 3 METHOD

Our FGMOL (Figure 2) establishes a unified framework for SBDD by first injecting functional interaction priors into protein atomic representations (Section 3.1), then constructing a structure-function field through motif-level alignment (Section 3.2), and guiding molecular generation via posterior flows modeled by a Bayesian Flow Network (Section 3.3). In this way, FGMOL transforms traditional structure-based generation into structure-function guided design, forming a closed-loop system that tightly couples structure, functional interaction, and generation.

### 3.1 INTERACTION-AWARE EMBEDDING

The key challenge in SBDD is not merely identifying where a ligand binds, but understanding why it binds. Although protein structures are available, their underlying functional interactions—such as hydrogen bonding, hydrophobic contacts, and $\pi-\pi$ stacking—are often neglected. To address this, FGMOL encodes functional interaction priors into protein representations and models protein–ligand complexes using an SE(3)-equivariant NN. This integration provides a function-aware structural representation as the foundation for subsequent alignment and generation.

**Functional Interaction Encoding** To inject functional priors into protein atom embeddings, we annotate each protein atom $i$ with a multi-hot vector $\mathbf{a}_i \in \{0, 1\}^{N_a}$ (Salentin et al., 2015), where $N_a$ denotes the number of predefined interaction types. These interactions types include anion, cation, hydrogen bond donor/acceptor, aromatic, hydrophobic, and non-interacting (see Appendix A). Each

entry $a_{i,j}$ in the vector is defined as:

$$a_{i,j} = \begin{cases} 1 & \text{if atom } i \text{ belongs to class } j, \\ 0 & \text{otherwise.} \end{cases} \tag{6}$$

We concatenate $\mathbf{a}_i$ with the initial protein atom embedding $\mathbf{h}'_{p,i}$, followed by a linear projection:

$$\mathbf{h}^0_{p,i} = \text{Linear}([\mathbf{h}'_{p,i}, \mathbf{a}_i]), \tag{7}$$

where $[\cdot, \cdot]$ denotes concatenation along the last dimension. The resulting vector $\mathbf{h}^0_{p,i}$ encodes both structural and functional information, yielding an interaction-aware protein representation.

**SE(3)-equivariant NN Modeling** To ensure that ligand generation is guided not only by geometric constraints but also by the functional preferences of the binding site, we inject interaction-aware features into protein atoms and construct a unified protein–ligand graph, where each node (protein or ligand) is initialized with its embedding, forming the full feature matrix $\mathbf{h}^0_G = [\mathbf{h}^0_m, \mathbf{h}^0_p]$. We then employ an SE(3)-equivariant neural network (Fuchs et al., 2020; Thomas et al., 2018) to propagate information across the 3D structure, enabling the ligand conformation to evolve toward geometrically plausible and biochemically aligned configurations. At each message-passing layer $l$, both hidden features and coordinates are updated as:

$$\mathbf{h}^{l+1}_{G,i} = \mathbf{h}^l_{G,i} + \sum_{j \in \mathcal{N}_G(i)} f^l_h \left( \|\Delta\mathbf{x}^l_{G,ij}\|, \mathbf{h}^l_{G,i}, \mathbf{h}^l_{G,j}, \mathbf{e}_{G,ij} \right),$$

$$\mathbf{x}^{l+1}_{G,i} = \mathbf{x}^l_{G,i} + \mathbb{I}_{\text{mol}} \cdot \sum_{j \in \mathcal{N}_G(i)} \Delta\mathbf{x}^l_{G,ij} \cdot f^l_x \left( \|\Delta\mathbf{x}^l_{G,ij}\|, \mathbf{h}^{l+1}_{G,i}, \mathbf{h}^{l+1}_{G,j}, \mathbf{e}_{G,ij} \right), \tag{8}$$

where $\Delta\mathbf{x}^l_{G,ij} = \mathbf{x}^l_{G,i} - \mathbf{x}^l_{G,j}$ is the relative position vector, and $\mathcal{N}_G(i)$ denotes the $k$-nearest neighbors of atom $i$. The edge feature $\mathbf{e}_{G,ij}$ encodes the edge type—whether it connects two protein atoms, two ligand atoms, or a cross-complex pair. $\mathbb{I}_{\text{mol}}$ masks updates to ligand atoms only, with protein positions held constant.

## 3.2 Interaction-Informed Motif Alignment

Although atom-level interactions are critical, effective molecular binding often arises from pharmacophoric atom groups (i.e., motifs). FGMOL captures these pharmacophore-like interactions by clustering atoms into higher-order semantic motifs and aligning them across ligand and protein structures. This design embeds a function-aware inductive bias, advancing SBDD from geometry-driven fitting to biologically grounded, interaction-informed ligand generation.

**Differentiable Pooling** We first compute soft cluster assignments for ligand and protein atoms to group them with similar interaction semantics (Ying et al., 2018). The soft cluster assignment matrices $\mathbf{s}_m \in \mathbb{R}^{|v_m| \times c_m}$ and $\mathbf{s}_p \in \mathbb{R}^{|v_p| \times c_p}$ are given by:

$$\mathbf{s}_m = \text{softmax}\left(\text{GNN}_{\mathbf{w}_m}\right)\left(\mathbf{h}^L_m, e_m\right), \quad \mathbf{s}_p = \text{softmax}\left(\text{GNN}_{\mathbf{w}_p}\right)\left(\mathbf{h}^L_p, e_p\right), \tag{9}$$

where $\mathbf{h}^L_m \in \mathbb{R}^{|v_m| \times d}$ and $\mathbf{h}^L_p \in \mathbb{R}^{|v_p| \times d}$ denote the ligand and protein atom embeddings at the $L^{\text{th}}$ layer, respectively. $e_m$ and $e_p$ are the corresponding intra-molecular edge features. $c_m$ and $c_p$ denote the numbers of clusters for the ligand and protein, respectively. Cluster-level representations $\mathbf{z}_m \in \mathbb{R}^{c_m \times d}$ and $\mathbf{z}_p \in \mathbb{R}^{c_p \times d}$ are then obtained by aggregating the atom embeddings with the assignment matrices:

$$\mathbf{z}_m = \mathbf{s}^T_m \mathbf{h}^L_m, \quad \mathbf{z}_p = \mathbf{s}^T_p \mathbf{h}^L_p. \tag{10}$$

These clusters serve as dynamically learned pharmacophore-like motifs, providing flexible, data-driven guidance that goes beyond purely geometric constraints. In addition, to promote sharper and more semantically coherent motif information, we impose an entropy regularization term on the assignment distributions. It is defined as: $L_e = \frac{1}{b} \sum_{i=1}^b (w_{em} H(\mathbf{s}_{m,i}) + w_{ep} H(\mathbf{s}_{p,i}))$, where $b$ denotes the batch size and $H(\cdot)$ is the Shannon entropy (Shannon, 1948).

**Cross-Attention Alignment**    To align ligand and protein clusters, we compute a similarity matrix as:

$$\mathbf{A}_{\text{align}} = \exp\left(\frac{\mathbf{z}_m \mathbf{z}_p^T}{\tau}\right), \tag{11}$$

where $\tau$ is a temperature parameter controlling sharpness. We apply Sinkhorn normalization (Cuturi, 2013) by alternating row and column normalization to obtain a doubly stochastic matrix $\mathbf{A}_{\text{final}}$:

$$\mathbf{A}[:,j] = \frac{\mathbf{A}[:,j]}{\sum_i \mathbf{A}(i,j)} \quad \forall j, \quad \mathbf{A}[i,:] = \frac{\mathbf{A}[i,:]}{\sum_j \mathbf{A}(i,j)} \quad \forall i. \tag{12}$$

Here, matrix indices are omitted for clarity. After $n$ iterations, we refine the ligand cluster representations by incorporating aligned protein context:

$$\mathbf{z}'_m = \mathbf{A}_{\text{final}} \mathbf{z}_p + \mathbf{z}_m. \tag{13}$$

This refinement injects protein-derived interaction context into ligand motifs while preserving their semantics, facilitating cross-target generalization. Using $\mathbf{z}'_m$, we apply a cross-attention mechanism to enrich atom-level ligand representations:

$$\mathbf{h}'^L_m = \text{softmax}\left(\frac{\mathbf{W}_Q \mathbf{h}^L_m (\mathbf{W}_K \mathbf{z}'_m)^\top}{\sqrt{d}}\right)(\mathbf{W}_V \mathbf{z}'_m), \tag{14}$$

where $\mathbf{W}_Q$, $\mathbf{W}_K$, and $\mathbf{W}_V$ are learnable parameters. $\mathbf{h}'^L_m$ and $\mathbf{x}^L_m$ are used to predict $\hat{\mathbf{v}}_m$ and $\hat{\mathbf{x}}_m$, respectively. Further details are available in Appendix C.4.

Overall, interaction-informed motif alignment yields a structured and functionally grounded representation space, bridging the gap between semantic recognition and generative modeling. These enriched embeddings lay the foundation for interaction-aware ligand generation, which we describe next.

## 3.3 INTERACTION-GUIDED GENERATION WITH BFN

Unlike diffusion models that perform local denoising in the observation space, FGMOL employs a Bayesian Flow Network (BFN) (Graves et al., 2023) to model posterior flows in the continuous parameter space. This enables generation trajectories to follow distributions informed by functional interaction priors. Through trajectory-level optimization, BFN embeds these priors throughout the generation process, ensuring that ligands progressively conform to biologically meaningful interaction patterns. Modeling in the continuous parameter space also mitigates the instability caused by discrete sampling noise, thereby enhancing generation robustness.

**Function-Aware Sampling in Parameter Space**    During sampling, the learned model $\Phi$ is used for molecular generation. Following Qu et al. (2024), we perform sampling entirely in continuous parameter space to avoid the instability and representational mismatch that arise from switching between continuous (coordinate) and discrete (type) representations.

The sampling first predicts intermediate parameters $\hat{\mathbf{m}} = [\hat{\mathbf{x}}, \hat{\mathbf{v}}]$, then updates them via a learned flow distribution $p_F$ (Eq. 4). Specifically, defining $\gamma(t) \overset{\text{def}}{=} \frac{\beta(t)}{1-\beta(t)}$, the update step for coordinates is:

$$p_F(\boldsymbol{\mu} \mid \hat{\mathbf{x}}_m, \mathbf{p}, \mathbf{a}; t) = \mathcal{N}(\boldsymbol{\mu} \mid \gamma(t)\hat{\mathbf{x}}, \gamma(t)(1-\gamma(t))\mathbf{I}), \tag{15}$$

where $\hat{\mathbf{x}}$ is the predicted coordinate at timestep $t$. For atom types:

$$p_F\left(\boldsymbol{\theta}^v \mid \hat{\mathbf{v}}_m, \mathbf{p}, \mathbf{a}; t\right) = \underset{\mathcal{N}(\mathbf{y}^v \mid \beta(t)(K\mathbf{e}_{\hat{\mathbf{v}}} - \mathbf{1}), \beta(t)K\mathbf{I})}{\mathbb{E}} \delta\left(\boldsymbol{\theta}^v - \text{softmax}\left(\mathbf{y}^v\right)\right). \tag{16}$$

This variance-controlled sampling strategy yields smooth, function-aware generation trajectories by operating entirely in the continuous parameter space. Instead of injecting new noise at each step, the model uses the predicted $\hat{\mathbf{m}} = [\hat{\mathbf{x}}, \hat{\mathbf{v}}]$ to update parameters for the next step, with $\hat{\mathbf{v}}$ kept in its continuous categorical form without discretization. These parameters are propagated via $\boldsymbol{\theta}_i = h(\boldsymbol{\theta}_{i-1}, \mathbf{y}, \alpha)$. This design transforms the generation process into a deterministic yet flexible trajectory through latent space, enabling fine-grained guidance from both structural geometry and functional interaction priors. Sampling details are provided in Algorithm 1 in Appendix C.5.

Table 1: Comparison of reference and generated molecules by our model and other baselines across key metrics. ↑/↓ indicate whether higher or lower values are preferred. Top-2 results are marked in **bold** and underlined, respectively. Note: HA = High Affinity, SE = Strain Energy, Div = Diversity.

| Methods | Vina Score (↓) | | Vina Min (↓) | | Vina Dock (↓) | | HA (↑) | SE (↓) | | | QED (↑) | SA (↑) | Diversity (↑) | Success Rate (↑) |
|---|---|---|---|---|---|---|---|---|---|---|---|---|---|---|
| | Avg. | Med. | Avg. | Med. | Avg. | Med. | Avg. | 25% | 50% | 75% | Avg. | Avg. | Avg. | Avg. |
| Reference | -6.36 | -6.46 | -6.71 | -6.49 | -7.45 | -7.26 | - | 34 | 107 | 196 | 0.48 | 0.73 | - | 25.0% |
| LiGAN | - | - | - | - | -6.33 | -6.20 | 21.1% | - | - | - | 0.39 | 0.59 | 0.66 | 3.9% |
| AR | -5.75 | -5.64 | -6.18 | -5.88 | -6.75 | -6.62 | 42.26% | 259 | 595 | 2286 | 0.51 | 0.63 | 0.70 | 7.1% |
| Pocket2Mol | -5.14 | -4.70 | -6.42 | -5.82 | -7.15 | -6.79 | 46.66% | 102 | 189 | 374 | **0.52** | **0.74** | 0.69 | 24.4% |
| Ours-small | **-5.94** | **-5.82** | **-6.61** | **-6.28** | **-7.41** | **-6.92** | **48.94%** | 94 | 169 | 347 | 0.50 | 0.70 | 0.72 | **26.1%** |
| FLAG[2] | 16.48 | 4.53 | 1.21 | -4.04 | -5.63 | -6.61 | 27.16% | 143 | 396 | 1164 | **0.70** | 0.49 | 0.70 | 14.1% |
| TargetDiff | -5.47 | -6.30 | -6.64 | -6.83 | -7.80 | -7.91 | 56.42% | 369 | 1243 | 13871 | 0.48 | 0.58 | 0.72 | 10.5% |
| DecompDiff | -5.19 | -5.27 | -6.03 | -6.00 | -7.03 | -7.16 | 48.15% | 115 | 421 | 1424 | 0.45 | 0.51 | 0.73 | 14.9% |
| MolCRAFT | -6.59 | -7.04 | -7.27 | -7.26 | -7.92 | -8.01 | 61.74% | 83 | 195 | 510 | 0.50 | 0.69 | 0.72 | 26.8% |
| FGMOL (Ours) | **-7.02** | **-7.10** | **-7.86** | **-7.59** | **-8.51** | **-8.52** | **62.07%** | **64** | **168** | **403** | 0.49 | 0.70 | **0.79** | **28.6%** |

# 4 EXPERIMENTS

## 4.1 EXPERIMENTAL SETTINGS

**Datasets and Baselines** Following Guan et al. (2023a) and Qu et al. (2024), we utilize the CrossDocked2020 dataset (Francoeur et al., 2020) as the benchmark for training and evaluation of our FGMOL model. Data preprocessing and train-test splitting follow Luo et al. (2021), wherein the original 22.5 million docking complexes are filtered to retain high-quality conformations (RMSD < 1 Å from crystal structures) and structurally diverse proteins (sequence identity < 30%), thereby ensuring sufficient diversity. From this curated dataset, we select 100,000 complexes for training and 100 distinct proteins as test pockets for downstream evaluation.

We compare our model with several generation-based baselines, including autoregressive models such as **LiGAN** (Ragoza et al., 2022), **AR** (Luo et al., 2021), and **Pocket2Mol** (Peng et al., 2022), which generate ligands sequentially. We also evaluate diffusion-base approaches, including **TargetDiff** (Guan et al., 2023a) and **DecompDiff** (Guan et al., 2023b), and the flow-based model **MolCRAFT** (Qu et al., 2024), which performs generation via continuous parameter space. More baseline results can be found in Table 4 of Appendix F.

**Evaluation** We evaluate generated molecules from five key perspectives: **(i) Molecular Structures**. We evaluate the structural quality by computing the Jensen-Shannon divergences (**JSD**) between atom and bond distance distributions of generated versus reference molecules—lower values indicating higher geometric similarity. To further quantify stability, we report the $25^{th}$, $50^{th}$, and $75^{th}$ percentiles of strain energy across generated conformations, where lower energies indicate greater physical plausibility. **(ii) Target Binding Affinity**. Following previous work (Luo et al., 2021; Ragoza et al., 2022), we employ four AutoDock (Eberhardt et al., 2021) Vina-based metrics: **Vina Score** (raw affinity), **Vina Min** (scoring after local relaxation), **Vina Dock** (re-docking based affinity), and **High Affinity** (fraction exceeding reference ligands). **(iii) Molecular Properties**. To assess chemical quality, we adopt three metrics: **QED** (drug-likeness), **SA** (synthetic accessibility), and **Diversity** (capturing the average pairwise dissimilarity among generated ligands for each target). **(iv) Overall.** We report the **Success Rate**—the proportion of molecules satisfying QED > 0.25, SA > 0.59, and Vina Dock < -8.18—to quantify both chemical quality and binding performance. **(v) Sample Efficiency.** We report the average number of samples generated per second to enable practical comparison.

## 4.2 MAIN RESULTS

Table 1 includes two variants of FGMOL—FGMOL-small and FGMOL—distinguished by their sampling strategies controlling molecule size (Qu et al., 2024).

**Binding Affinity** FGMOL achieves state-of-the-art performance in terms of Vina Score (-7.02), Vina Min (-7.86), and Vina Dock (-8.51) in Table 1, significantly outperforming previous best results from MolCRAFT (-7.92) and DecompDiff (-7.03). Notably, FGMOL achieves the highest proportion of high-affinity molecules (62.07%), supporting the efficacy of its interaction-aware design.

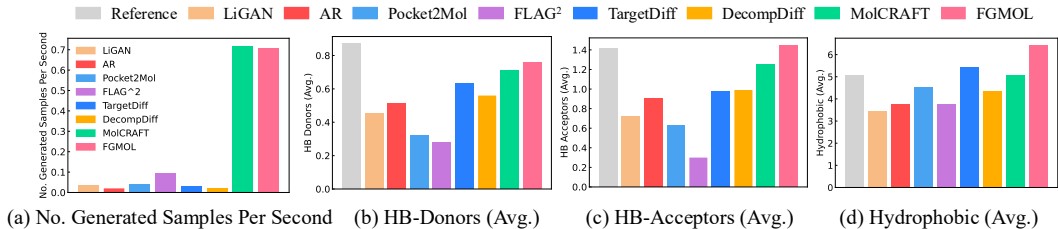

(a) No. Generated Samples Per Second  (b) HB-Donors (Avg.)  (c) HB-Acceptors (Avg.)  (d) Hydrophobic (Avg.)

Figure 3: Computational efficiency and interaction analysis. (a) Generation efficiency as samples per second. (b–d) Average counts of key molecular interactions for SBDD models.

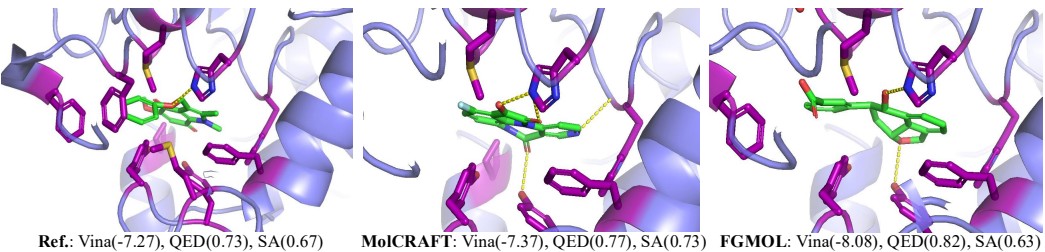

**Ref.**: Vina(-7.27), QED(0.73), SA(0.67)  **MolCRAFT**: Vina(-7.37), QED(0.77), SA(0.73)  **FGMOL**: Vina(-8.08), QED(0.82), SA(0.63)

Figure 4: Reference and generate ligands of MolCRAFT (Qu et al., 2024) and our FGMOL for 1gg5. We report Vina score, QED, and SA for each molecule. Yellow dashed lines represent hydrogen bonds.

**Structural Stability**   In Table 1, FGMOL substantially lowers strain energy across the $25^{th}$, $50^{th}$, and $75^{th}$ percentiles (64, 168, and 403, respectively), outperforming strong baselines like Pocket2Mol and MolCRAFT. This highlights the strength of BFN in modeling realistic conformations and the advantage of motif-level alignment in reducing local strain. In addition, Table 2 shows that FGMOL achieves the best overall conformation quality, demonstrating its ability to capture physically realistic geometries. Figure 3b–d and Figure 4 further show that our generated ligands match or even surpass the reference in terms of interaction count, underscoring their ability to retain essential interactions.

**Molecular Quality**   Regarding molecular property metrics, FGMOL achieves the highest synthetic accessibility score (0.70), indicating that the generated molecules are not only functional but also readily synthesizable. Its QED score (0.49) is competitive with other top-performing methods, while its diversity score (0.79) is the highest among all evaluated models, highlighting FGMOL is capable of generating chemically diverse candidates for a wide range of protein targets.

Table 2: Summary of molecular conformation results. (↓) indicates smaller is better. Top 2 results are highlighted with **bold** text and underlined text. JSD values reflect mean distributional differences between generated and reference geometries.

| Methods | Length (↓) Avg. JSD | Angle (↓) Avg. JSD | Torsion (↓) Avg. JSD |
|---|---|---|---|
| LiGAN | 0.638 | 0.602 | - |
| AR | 0.554 | 0.507 | 0.552 |
| Pocket2Mol | 0.485 | 0.482 | 0.459 |
| FLAG | 0.511 | 0.406 | **0.270** |
| TargetDiff | 0.382 | 0.435 | 0.400 |
| DecompDiff | 0.348 | 0.412 | 0.317 |
| MolCRAFT | 0.319 | 0.379 | 0.300 |
| Ours | **0.298** | **0.356** | 0.284 |

**Overall**   FGMOL achieves the highest overall Success Rate (28.6%) under the composite evaluation criterion (QED > 0.25, SA > 0.59, Vina Dock < -8.18), outperforming MolCRAFT (26.8%) and Pocket2Mol (24.4%). This demonstrates FGMOL's ability to generate ligands that simultaneously satisfy pharmacological, chemical, and structural requirements, setting a new benchmark for interaction-aware molecular generation.

**Sample Efficiency**   Figure 3a shows that FGMOL achieves a significantly higher sample generation speed compared to all baseline methods, with over 0.7 samples per second, matching the efficiency of MolCRAFT. In contrast, diffusion- and autoregressive-based baselines such as TargetDiff, DecompDiff, and AR exhibit much lower throughput (typically under 0.1 samples per second), highlighting the efficiency advantage of our continuous-space, BFN-based generation strategy.

Table 3: Effect of different modules on the generation performance of FGMOL. The best results are marked in **bold** and runner-ups are underline. The original FGMOL is incorporated for comparison.

| Methods | Vina Score (↓) | | Vina Min (↓) | | Vina Dock (↓) | | SE (↓) | | | SA (↑) |
|---|---|---|---|---|---|---|---|---|---|---|
| | Avg. | Med. | Avg. | Med. | Avg. | Med. | 25% | 50% | 75% | Avg. |
| Exp0 (Baseline) | -6.59 | -7.04 | -7.27 | -7.26 | -7.92 | -8.01 | 83 | 195 | 510 | 0.69 |
| Exp1 (+ Functional annotations) | -6.74 | -6.06 | -7.24 | -7.28 | -7.89 | -8.03 | 80 | 196 | 521 | **0.71** |
| Exp2 (+ Differentiable pooling and cross-attention ) | -6.85 | -7.01 | -7.42 | -7.56 | -8.08 | -8.32 | 76 | 174 | 442 | 0.69 |
| Exp3 (+ Sinkhorn alignment) | -6.98 | **-7.12** | -7.74 | -7.52 | -8.34 | -8.47 | 79 | 179 | 452 | 0.70 |
| Exp4 (+ Entropy regularization) | **-7.02** | -7.10 | **-7.86** | **-7.59** | **-8.51** | **-8.52** | **64** | **168** | **403** | 0.70 |

## 4.3 ABLATION STUDIES

Table 3 presents ablation experiments conducted to evaluate the individual contributions of each component in FGMOL. The base model (Exp0), which lacks interaction modeling, demonstrates the weakest overall performance. Introducing functional annotations (Exp1) improves the synthetic accessibility (SA) of the generated molecules. Incorporating motif-level pooling and attention mechanisms (Exp2) further enhances energy-related metrics, reflecting improved structural plausibility. The addition of Sinkhorn alignment (Exp3) leads to notable gains in both docking and binding scores, underscoring the importance of accurate motif-level functional matching. Finally, applying entropy regularization (Exp4) results in the best overall performance across all evaluation metrics, confirming that each module incrementally contributes to generation quality. Overall, FGMOL achieves an optimal balance among binding affinity, structural precision, and synthetic feasibility. More ablation results are provided in Table 12 of Appendix F.

## 4.4 GENERALIZATION OF FGMOL

Table 13 presents a comprehensive comparison on the PoseBusters benchmark (Qiu et al., 2025) under out-of-distribution (OOD) scenarios. The results demonstrate that FGMOL achieves the best overall performance, attaining the highest PB-Valid score (82.10%), superior docking results across all metrics, and the strongest structural accuracy, with 58.4% of generated poses reaching scRMSD<2Å. Beyond structural fidelity, FGMOL also preserves favorable chemical properties by maintaining competitive QED and SA values, while achieving the highest molecular diversity (0.79) with relatively compact molecular size. These results collectively indicate that FGMOL delivers a balanced improvement across structural, energetic, and chemical dimensions, establishing it as a robust and effective framework for molecular pose generation under OOD conditions.

## 4.5 HYPERPARAMETER ANALYSIS

We investigate the impact of five key hyperparameters on the performance of FGMOL, including the number of ligand clusters ($c_m$), the number of protein clusters ($c_p$), the entropy loss weight of the ligand cluster alignment matrix ($w_{em}$), and the entropy loss weight of the protein cluster alignment matrix ($w_{ep}$). As shown in Tables 5 and 6 in the Appendix F, the best performance is achieved when $c_m$ and $c_p$ are set to 6 and 32, respectively. Additionally, Table 7 demonstrates that setting $w_{em}$ to 0.2 yields optimal results, and Table 8 shows that $w_{ep} = 0.2$ also leads to the best performance.

## 5 CONCLUSION

In this work, we propose FGMOL, a unified framework for structure-based drug design (SBDD) that integrates protein structure, functional interactions, and ligand generation within a continuous parameter space. By incorporating biochemical priors, differentiable motif alignment, and Bayesian flow-based generation, FGMOL is able to capture both geometric and functional compatibility. Experiments on CrossDocked2020 show that FGMOL achieves state-of-the-art results across multiple recognized metrics. This work underscores the effectiveness of combining structural and biochemical information for interaction-aware molecular design.

**Limitation** Despite its effectiveness, our FGMOL has two main limitations. First, it depends on atom-level interaction annotations, which are often unavailable in the case of sparsely annotated proteins. Second, it assumes a rigid protein structure, ignoring conformational flexibility and protein–ligand adaptation. Consequently, it cannot capture induced fit and is less effective in dynamic binding scenarios. To address these issues, we plan to develop a model that jointly performs interaction type classification and ligand generation under flexible protein conformations.

ETHICS STATEMENT

This paper proposes a new method, FGMOL , a unified framework for structure-based drug design that integrates protein structure, functional interactions, and ligand generation within a continuous parameter space. To the best of our knowledge, no ethical concerns arise beyond those commonly associated with research in this field.

REPRODUCIBILITY STATEMENT

We have described the model architecture, training procedure, and evaluation protocols in detail within the paper to ensure reproducibility. Upon acceptance, we will make all source data and the full implementation publicly available.

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

## A   PROTEIN-LIGAND INTERACTION ANALYSIS

A key innovation of FGMOL lies in its explicit modeling of protein-ligand functional interactions, moving beyond traditional geometry-based pocket fitting. To encode biochemical priors, we annotate each protein atom with a discrete interaction role. These roles include **hydrogen bond donor**, **hydrogen bond acceptor**, **hydrophobic**, **aromatic**, **anion**, **cation**, and **non-interacting**. Below, we elaborate on the four key interaction types that critically govern ligand binding and structural complementarity.

**Hydrogen Bonds (H-bonds).**    Hydrogen bonding is a directional interaction between a hydrogen donor (typically –OH or –NH groups) and an acceptor (commonly O or N atoms with lone pairs) (McDonald & Thornton, 1994). These interactions are crucial for anchoring ligands within the binding pocket and contribute to binding specificity (Bissantz et al., 2010). Following structural bioinformatics conventions (e.g., PLIP (Adasme et al., 2021)), we assign H-bonds when the donor–acceptor distance is below 3.5 Å and the donor–hydrogen–acceptor angle exceeds $120°$. Both donors and acceptors are labeled on protein atoms using rule-based heuristics and residue-level annotations (Schmidtke & Barril, 2010).

**Hydrophobic Interactions.**    Hydrophobic effects arise when nonpolar atoms cluster together to avoid water exposure, thereby stabilizing the protein-ligand complex (Chothia, 1974; Dill, 1990). In our implementation, hydrophobic atoms are defined as aliphatic carbons not bonded to polar groups Böhm (1992). We also mark spatially proximal nonpolar atoms ($< 4.0$ Å apart) across the protein-ligand interface as engaging in hydrophobic interactions (Laskowski et al., 1996; Adasme et al., 2021). These regions are particularly important for shape complementarity and entropic stabilization .

**Aromatic and $\pi$–$\pi$ Interactions.**    Aromatic interactions involve $\pi$–electron stacking between conjugated ring systems. These interactions are prevalent in drug-like molecules and often contribute significantly to binding affinity via van der Waals and quadrupole effects Meyer et al. (2003). We identify aromatic atoms as those within six-membered planar ring systems satisfying Hückel's rule. $\pi$–$\pi$ stacking is detected when two aromatic systems are within 5.0 Å and the dihedral angle between ring planes is below $30°$ (face-to-face) or between $60°$ and $120°$ (T-shaped) McGaughey et al. (1998).

**Ionic Interactions (Salt Bridges).**    Ionic interactions occur between oppositely charged groups—commonly between acidic side chains (e.g., Asp, Glu) and basic residues (e.g., Lys, Arg) or charged ligand groups. We label protein atoms as cations or anions based on residue identity and formal charge assignments. Salt bridges are defined when the distance between ionizable atoms is less than 4.0 Å (Kumar & Nussinov, 2000). These interactions are critical for long-range electrostatic steering and early-stage binding recognition (Honig & Nicholls, 1995).

Each protein atom is represented by a one-hot interaction vector $\mathbf{a}_i \in \{0, 1\}^{N_a}$, where $N_a = 7$. These annotations are concatenated with the atom embeddings and projected through a linear layer to yield function-aware atomic representations used in SE(3)-equivariant message passing.

# B    RELATED WORK

**Structure-Based Drug Design**    Structure-based drug design (SBDD) (Zhang et al., 2023d; Du et al., 2024; Gao et al., 2025; Lin et al., 2025) aims to generate high-affinity 3D ligand molecules based on the target protein pocket. LiGAN (Ragoza et al., 2022) first voxelizes molecules into atomic-density grids and integrates a C-VAE (Pagnoni et al., 2018) for ligand generation. Subsequently, autoregressive models have been developed to generate molecules at the atom level (Liu et al., 2022; Luo & Ji, 2022; Zhang et al., 2023b;a) or functional group level (Green et al., 2021; Powers et al., 2022). For example, Pocket2Mol (Peng et al., 2022) introduces E(3)-equivariant graph neural networks for joint atom-level generation. Additionally, FLAG (Zhang et al., 2023c) and DrugGPS (Zhang et al., 2023a) adopt a fragment-based generation strategy to generate molecules. However, these models often suffer from limited global context and error accumulation (Guan et al., 2023a). To address these issues, some studies (Schneuing et al., 2024; Qu et al.) have introduced diffusion models to advance SBDD. For instance, TargetDiff (Guan et al., 2023a) and DiffSBDD (Schneuing et al., 2024) combine SE(3)-equivariant diffusion models with non-autoregressive generation strategies for 3D molecule generation. Furthermore, Decompdiff (Guan et al., 2023a) and IPDiff (Huang et al., 2024b) incorporate prior knowledge (e.g., motif-based or interaction-based) into the SBDD task. Unlike prior geometry-focused approaches, FGMoL establishes a unified structure-function field by injecting explicit functional interaction priors and aligning protein-ligand motifs, thereby enabling molecular generation that is both function-aware and continuous in parameter space, Appendix B provides additional details.

**Protein-Ligand Interaction Modeling**    In the domain of protein-ligand interaction modeling, various approaches have been developed to enhance prediction and generation capabilities by integrating prior knowledge or leveraging existing models (Hassan et al., 2005; Du et al., 2016; Adams et al., 2025). 1) Prior Knowledge Utilization: Zhang & Chen (2022) combines Ligand-protein interaction fingerprint (LPI) (Deng et al., 2004; Bouysset & Fiorucci, 2021) constraints with an RNN architecture to maintain the consistency of ligand-protein binding; Interformer (Lai et al., 2024) improves model accuracy by explicitly modeling functional interactions between molecules through the use of Interaction-aware MDN, edge features, and Pose Energy calculation mechanisms; FeatureDock (Xue et al., 2025) refines prediction accuracy by extracting physicochemical features from protein pockets and optimizing scoring functions for virtual screening. 2) Model-Based Methods: PMINet (Huang et al., 2024a) and IPNET (Huang et al., 2024b) employ docking models to predict interaction data, such as affinity, to further enhance molecule generation. In contrast, our FGMOL focuses on the local microenvironment of protein-ligand interactions, using a pooling strategy to capture interaction-based clusters, thereby improving the modeling of complex molecular interactions.

## C    DETAILED FORMULATION OF BFN

### C.1    INTRODUCING PARAMETERS INTO THE DIFFUSION PROCESS

In classical diffusion models, the generative procedure consists of two principal stages:

- **Forward Diffusion:** A data sample $\mathbf{m}$ is progressively perturbed by injecting noise over multiple steps, ultimately resulting in a standard Gaussian variable $y_0 \sim \mathcal{N}(0, I)$. Each intermediate step follows the transition $y_i \sim p(y_i \mid \mathbf{m}; \alpha_i)$.

- **Reverse Denoising:** Starting from Gaussian noise $y_0$, the model reconstructs the data by iteratively removing noise conditioned on the target $\mathbf{p}$, using the transition $y_i \sim p(y_i \mid y_{i-1}, \mathbf{p}; t)$, eventually recovering the sample via $\mathbf{m} \sim p(\mathbf{m} \mid y_n, \mathbf{p}; n)$.

The key design challenges in diffusion models are twofold: determining how to inject noise conditioned on $\mathbf{m}$, and how to effectively denoise using $(y, \mathbf{p})$.

### C.2    VARIATIONAL LOWER BOUND AND PARAMETER SPACE MODELING

As proposed by Ho et al. (2020), the training objective is formulated through the variational lower bound (VLB):

$$-\log p_\theta(\mathbf{m} \mid \mathbf{p}) \leq \mathcal{L}_{\text{VLB}} = D_{\text{KL}}\left(q(y_0, \dots, y_n \mid \mathbf{m}, \mathbf{p}) \,\|\, p_\phi(y_0, \dots, y_n \mid \mathbf{p})\right) \tag{17}$$

To transition from sample space to parameter space, a sequence of latent variables $\theta_0, \dots, \theta_n$ is introduced to parameterize the intermediate representations. This allows the variational objective to be reformulated as:

$$\mathcal{L}_{\text{VLB}} = n \cdot \mathbb{E}_{i \sim \mathcal{U}(1,n),\, y_i, \theta_{i-1} \sim q}\left[D_{\text{KL}}\left(q(y_i \mid \mathbf{m}) \,\|\, p_\phi(y_i \mid \theta_{i-1}, \mathbf{p})\right)\right] \tag{18}$$

This formulation decomposes the KL divergence across the trajectory, enabling the model to learn a step-wise denoising process that aligns the target sample with a parameterized latent pathway.

### C.3    SENDER AND RECEIVER DISTRIBUTIONS IN BFN

Within the Bayesian Flow Network (BFN) framework, the diffusion dynamics are governed by two key components:

- **Sender Distribution:** Controls the forward noise injection process:

$$q(y_i \mid \mathbf{m}) = p_S(y_i \mid \mathbf{m}; \alpha_i) \tag{19}$$

- **Receiver Distribution:** Drives the reverse generation process, denoising $y_i$ conditioned on the latent state $\theta_{i-1}$ and the target $\mathbf{p}$.

The training objective is to align the receiver distribution $p_\phi$ with the forward trajectory induced by the sender. This alignment allows the model to alternately generate $y_i$ and latent variables $\theta_i$, thereby enabling an efficient, non-autoregressive, and target-aware generative mechanism.

## C.4 TRAINING WITH BFN

Building on the continuous nature of BFN, we inject Gaussian noise into both atomic coordinates and types to construct the sender distributions. Specifically, the noisy sender distribution for coordinates is defined as $p_S \left( \mathbf{y}^x \mid \mathbf{x}_M; \alpha \right) = \mathcal{N} \left( \mathbf{y}^x \mid \mathbf{x}_M, \alpha^{-1}\mathbf{I} \right)$, and for atom types as $p_S \left( \mathbf{y}^v \mid \mathbf{v}_M; \alpha' \right) = \mathcal{N} \left( \mathbf{y}^v \mid \alpha' \left( K\mathbf{e}_{\mathbf{v}_M} - \mathbf{1} \right), \alpha' K\mathbf{I} \right)$, where $\mathbf{e}_{\mathbf{v}_M} = \left[ \mathbf{e}_{\mathbf{v}_M^{(1)}}, \ldots, \mathbf{e}_{\mathbf{v}_M^{(N_M)}} \right] \in \mathbb{R}^{N_M \times K}$, and each $\mathbf{e}_j \in \mathbb{R}^K$ represents a one-hot vector encoding atom class $j$. Distinct noise schedules ($\alpha$ for coordinates and $\alpha'$ for types) are adopted to facilitate efficient joint training.

The receiver distributions are defined similarly. For coordinates: $p_R \left( \mathbf{y}^x \mid \boldsymbol{\theta}^x, \mathbf{p}, \mathbf{a}; t \right) = \mathcal{N} \left( \mathbf{y}^x \mid \boldsymbol{\Phi} \left( \boldsymbol{\theta}^x, \mathbf{p}, \mathbf{a}, t \right), \alpha^{-1}\mathbf{I} \right)$, where $\boldsymbol{\Phi}$ denotes the prediction network. For atom types:

$$p_R \left( \mathbf{y}^v \mid \boldsymbol{\theta}^v, \mathbf{p}, \mathbf{a}; t \right) = \left[ p_R \left( (\mathbf{y}^v)^{(d)} \mid \cdot \right) \right]_{d=1\ldots N}, \tag{20}$$

where $p_R \left( (\mathbf{y}^v)^{(d)} \mid \cdot \right) = \sum_k p_O^v(k \mid \cdot) p_S^v \left( (\mathbf{y}^v)^{(d)} \mid k; \alpha \right)$.

**Training Objective**  The overall training loss consists of two components: i) KL divergence between sender and receiver distributions for both coordinates and atom types, and ii) an entropy regularization term over the functional motif alignment matrix. For coordinates, the KL divergence loss is:

$$\begin{aligned} L_x^n &= D_{\mathrm{KL}} \left( \mathcal{N} \left( \mathbf{x}, \alpha_i^{-1} I \right) \| \mathcal{N} \left( \hat{\mathbf{x}} \left( \boldsymbol{\theta}_{i-1}, \mathbf{p}, \mathbf{a}, t \right), \alpha_i^{-1} I \right) \right) \\ &= \frac{\alpha_i}{2} \| \mathbf{x} - \hat{\mathbf{x}} \left( \boldsymbol{\theta}_{i-1}, \mathbf{p}, t \right) \|^2 \end{aligned} \tag{21}$$

where $\hat{x}$ denotes the predicted coordinate. For atom types, the loss is given by:

$$\begin{aligned} L_v^n = &\ln \mathcal{N} \left( \mathbf{y}^v \mid \alpha_i \left( K\mathbf{e}_{\mathbf{v}} - \mathbf{1} \right), \alpha_i KI \right) - \\ &\sum_{d=1}^{N_M} \ln \left( \sum_{k=1}^{K} p_O(k \mid \boldsymbol{\theta}; t) \mathcal{N} \left( .^{(d)} \mid \alpha_i \left( K\mathbf{e}_k - \mathbf{1} \right), \alpha_i KI \right) \right). \end{aligned} \tag{22}$$

## C.5 SAMPLING

---

**Algorithm 1** Sampling

---

1: **function** UPDATE($\hat{\mathbf{x}} \in \mathbb{R}^{3N}, \hat{\mathbf{v}} \in \mathbb{R}^{NK}, \beta(t), \beta'(t), t \in \mathbb{R}^+$)
2:     $\gamma \leftarrow \frac{\beta(t)}{1-\beta(t)}$
3:     $\mu \sim \mathcal{N}(\gamma \hat{\mathbf{x}}, \gamma(1-\gamma)\mathbf{I})$
4:     $y^v \sim \mathcal{N}(y^v \mid \beta'(t)(K\hat{\mathbf{v}} - \mathbf{1}), \beta'(t)K\mathbf{I})$
5:     $\theta^v \leftarrow [\mathrm{softmax}((y^v)^{(d)})]_{d=1,\ldots,N_M}$
6:     **return** $\mu, \theta^v$
7: **end function**
**Require:** Network $\Phi, \mathbf{p} \in \mathbb{R}^{N_P(3+D_P)}, \mathbf{a} \in \mathbb{R}^{N_P \times N_a}, N, n_M, K \in \mathbb{N}^+, \sigma_1, \beta_1 \in \mathbb{R}^+$
8: $\mu \leftarrow \mathbf{0}, \rho \leftarrow 1, \theta^v \leftarrow \left[ \frac{1}{K} \right]_{n_M \times K}$
9: **for** $i = 1$ **to** $N$ **do**
10:     $t \leftarrow \frac{i-1}{n}$
11:     $\hat{\mathbf{x}}, \hat{\mathbf{v}} \leftarrow p_O(\mu, \theta^v, \mathbf{p}, \mathbf{a}, t)$
12:     $\mu, \theta^v \leftarrow \mathrm{UPDATE}(\hat{\mathbf{x}}, \hat{\mathbf{v}}, \sigma_1, \beta_1, t)$
13: **end for**
14: $\hat{\mathbf{x}}, p_O^v(\hat{\mathbf{v}} \mid \theta^v, \mathbf{p}, \mathbf{a}; 1) \leftarrow p_O(\mu, \theta^v, \mathbf{p}, \mathbf{a}, 1)$
15: $\hat{\mathbf{v}} \sim p_O^v(\hat{\mathbf{v}} \mid \theta^v, \mathbf{p}; 1)$
16: **return** $[\hat{\mathbf{x}}, \hat{\mathbf{v}}]$

---

The sampling procedure described in Algorithm 1 outlines an iterative strategy for generating structured molecular outputs conditioned on protein features. The algorithm maintains two key representations during the sampling trajectory: the 3D coordinates $\mu \in \mathbb{R}^{3N}$ representing atom positions, and a set of categorical distributions $\theta^v \in \mathbb{R}^{n_M \times K}$ over discrete latent states for vector-valued features.

At each iteration, the algorithm computes a normalized time step $t$, which controls the annealing schedule via time-dependent coefficients $\beta(t)$ and $\beta'(t)$. These coefficients modulate the variance of the Gaussian distributions used to update the atomic positions and latent vector assignments. The update function employs a Gaussian reparameterization centered at $\gamma \hat{x}$ for positional updates, and applies a softmax transformation to latent logits $y^v$ for updating $\theta^v$. The forward model $p_O(\cdot)$, parameterized by learned network components, provides conditional predictions based on current estimates of $\mu$ and $\theta^v$.

The algorithm proceeds for $N$ denoising steps, progressively refining $\mu$ and $\theta^v$ toward a structured sample. At the final step, discrete vector assignments $\hat{\mathbf{v}}$ are sampled from the predicted posterior distribution, yielding a complete sample $[\hat{\mathbf{x}}, \hat{\mathbf{v}}]$. This procedure enables controlled, conditional generation within a differentiable probabilistic framework, integrating geometric and categorical aspects of molecular design.

# D  PROOF OF SE(3) INVARIANT OBJECTIVE AND SE(3) EQUIVARIANT SAMPLING PROCESS

## D.1  SE(3)-EQUIVARIANCE AND TRANSFORMATION-INVARIANT OBJECTIVES

In molecular modeling, faithfully capturing the geometric invariances of 3D structures is crucial. To this end, we enforce invariance under the Special Euclidean group SE(3), which encompasses all possible rigid-body transformations (i.e., translations and rotations) in three-dimensional space (Satorras et al., 2021; Xu et al., 2022; Hoogeboom et al., 2022). An SE(3) transformation $T_g$ can be expressed as $T_g(\mathbf{x}) = \mathbf{R}\mathbf{x} + \mathbf{b}$, where $\mathbf{R} \in \mathbb{R}^{3 \times 3}$ is a rotation matrix and $\mathbf{b} \in \mathbb{R}^3$ is a translation vector.

To eliminate translational degrees of freedom, we follow the center-of-mass (CoM) normalization strategy described in Guan et al. (2023a). Specifically, given a protein representation $\mathbf{x}_P \in \mathbb{R}^{3N_P}$, we transform it to $\tilde{\mathbf{x}}_P = \mathbf{Q}\mathbf{x}_P$, where the centering matrix $\mathbf{Q}$ is defined as $\mathbf{I}_3 \otimes \left( \mathbf{I}_{N_P} - \frac{1}{N_P} \mathbf{1}_{N_P} \mathbf{1}_{N_P}^\top \right)$. Analogously, molecular and latent representations $\tilde{\mathbf{m}}, \tilde{\boldsymbol{\mu}}$ are also centered, such that under any rigid-body transformation $T_g$, we have:
$$T_g(\tilde{\mathbf{m}}) = \mathbf{R}\tilde{\mathbf{m}}, \quad T_g(\tilde{\boldsymbol{\mu}}) = \mathbf{R}\tilde{\boldsymbol{\mu}}, \quad T_g(\tilde{\mathbf{p}}) = \mathbf{R}\tilde{\mathbf{p}}.$$
Hereafter, we omit tildes for notational simplicity.

To illustrate the model's invariance behavior, we simplify the discrete-time update from Algorithm 1 as follows:
$$\boldsymbol{\mu} = \gamma \mathbf{x}_M + \gamma(1-\gamma)\boldsymbol{\epsilon}, \tag{23}$$
$$\hat{\mathbf{x}} = \Phi(\boldsymbol{\mu}, \mathbf{x}_P), \tag{24}$$
$$\mathcal{L}_x^n(\mathbf{x}_M, \mathbf{x}_P) = \text{const} \cdot \|\mathbf{x}_M - \hat{\mathbf{x}}\|^2. \tag{25}$$

Assuming $\boldsymbol{\epsilon}$ is sampled from an isotropic Gaussian distribution, any rotated version $\boldsymbol{\epsilon}' = \mathbf{R}\boldsymbol{\epsilon}$ remains identically distributed. Given that the denoising network $\Phi$ is SE(3)-equivariant, we obtain:
$$\Phi(T_g(\boldsymbol{\mu}, \mathbf{x}_P)) = \Phi(\mathbf{R}\boldsymbol{\mu}, \mathbf{R}\mathbf{x}_P) = \mathbf{R}\Phi(\boldsymbol{\mu}, \mathbf{x}_P) = T_g(\hat{\mathbf{x}}). \tag{26}$$

Consequently, the reconstruction loss remains unchanged under transformation:
$$\begin{aligned}
\|T_g(\mathbf{x}_M) - T_g(\hat{\mathbf{x}})\|^2 &= \|\mathbf{R}\mathbf{x}_M + \mathbf{b} - (\mathbf{R}\hat{\mathbf{x}} + \mathbf{b})\|^2 \\
&= \|\mathbf{R}(\mathbf{x}_M - \hat{\mathbf{x}})\|^2 \\
&= (\mathbf{x}_M - \hat{\mathbf{x}})^\top \mathbf{R}^\top \mathbf{R}(\mathbf{x}_M - \hat{\mathbf{x}}) \\
&= \|\mathbf{x}_M - \hat{\mathbf{x}}\|^2.
\end{aligned} \tag{27}$$

This confirms that the objective function is strictly invariant to SE(3) transformations of the input complex.

### D.2 GUARANTEEING EQUIVALENCE DURING SAMPLING

We now analyze the behavior of the iterative sampling procedure in Algorithm 2. To focus on 3D positional dynamics, auxiliary variables are omitted. We begin with:

$$\boldsymbol{\mu}_0 = \mathbf{0}, \quad \rho_0 = 1, \tag{28}$$

$$\mathbf{x}_i = \Phi(\boldsymbol{\mu}_{i-1}, \mathbf{x}_P), \tag{29}$$

$$\mathbf{y}_i^x = \mathbf{x}_i + \alpha_i \boldsymbol{\epsilon}, \tag{30}$$

$$\rho_i = \rho_{i-1} + \alpha_i, \tag{31}$$

$$\boldsymbol{\mu}_i = \frac{\rho_{i-1}\boldsymbol{\mu}_{i-1} + \alpha_i \mathbf{y}_i^x}{\rho_i}. \tag{32}$$

At the first iteration, since $\Phi$ is equivariant, we have:

$$T_g(\mathbf{x}_1) = T_g(\Phi(\boldsymbol{\mu}_0, \mathbf{x}_P)) = \Phi(T_g(\boldsymbol{\mu}_0, \mathbf{x}_P)) = \Phi(\boldsymbol{\mu}_0, T_g(\mathbf{x}_P)). \tag{33}$$

Hence, $\mathbf{x}_1$ is guaranteed to be SE(3)-equivariant with respect to $\mathbf{x}_P$ and $\boldsymbol{\mu}_0$.

Assume inductively that $T_g(\mathbf{x}_i) = \Phi(T_g(\boldsymbol{\mu}_{i-1}, \mathbf{x}_P))$. Then:

- $\mathbf{y}_i^x$ is equivariant to $\mathbf{x}_i$ by virtue of Gaussian noise being isotropic.
- $\boldsymbol{\mu}_i$ is computed from equivariant terms, hence it too is equivariant to $\mathbf{x}_P$.
- Consequently, $\mathbf{x}_{i+1}$ inherits this property from $\boldsymbol{\mu}_i$ and $\mathbf{x}_P$.

By induction, every intermediate state $\mathbf{x}_i$ throughout the trajectory remains equivariant to the initial reference frame. Ultimately, the final sample $\mathbf{x}_N$ also satisfies SE(3)-equivariance with respect to $\mathbf{x}_P$ and the initialized state $\boldsymbol{\mu}_0$.

## E IMPLEMENTATION DETAILS

### E.1 PARAMETERIZATION WITH SE(3) EQUIVARIANT NETWORK

To effectively model the spatial interactions between ligand atoms and protein pocket atoms, we build upon the SE(3)-equivariant framework and adopt PosNet3D (Guan et al., 2022) as our backbone. Given a protein-ligand complex, we first construct a graph $G = \langle V, E \rangle$ based on $k$-nearest neighbor (k-NN) search over atomic coordinates.

Within each layer $l$, atom-wise features $\mathbf{h}^l$ and 3D coordinates $\mathbf{x}^l$ are iteratively updated through two attention-based operations. The hidden representation is refined as:

$$\mathbf{h}_i^{l+1} = \mathbf{h}_i^l + \sum_{j \in \mathcal{N}_G(i)} \phi_h\left(d_{ij}, \mathbf{h}_i^l, \mathbf{h}_j^l, e_{ij}, t\right), \tag{34}$$

followed by a coordinate update computed from a learned displacement field:

$$\Delta\mathbf{x}_i = \sum_{j \in \mathcal{N}_G(i)} \left(\mathbf{x}_j^l - \mathbf{x}_i^l\right) \cdot \phi_x\left(d_{ij}, \mathbf{h}_i^{l+1}, \mathbf{h}_j^{l+1}, e_{ij}, t\right), \tag{35}$$

$$\mathbf{x}_i^{l+1} = \mathbf{x}_i^l + \Delta\mathbf{x}_i \cdot \mathbf{1}_{mol}, \tag{36}$$

where $\mathcal{N}_G(i)$ denotes the neighboring atoms of node $i$, and $d_{ij}$ is the Euclidean distance between atoms $i$ and $j$. The edge type $e_{ij}$ specifies the interaction type (protein–protein, ligand–ligand, or protein–ligand), and the binary mask $\mathbf{1}_{mol}$ ensures that only ligand coordinates are updated. Both attention modules $\phi_h$ and $\phi_x$ utilize $\mathbf{h}_i^l$ as the query, and $[\mathbf{h}_i^l, \mathbf{h}_j^l, e_{ij}]$ as keys and values.

At the input layer, the coordinate features are initialized by concatenating ligand embeddings and protein positions: $\mathbf{x}^0 = [\boldsymbol{\mu}, \mathbf{x}_P]$. The hidden state $\mathbf{h}^0$ is initialized through a linear projection of atom features and the time step $t$. At the output layer, the model produces predicted coordinates via $\hat{\mathbf{x}} = \Phi^x$. For discrete variables $\mathbf{v}^{(d)}$, predictions are obtained by applying softmax to the output logits:

$$\hat{\mathbf{v}}^{(d)} = \text{softmax}\left((\Phi^v)^{(d)}\right). \tag{37}$$

### E.2 FEATURIZATION

Following prior work (Qu et al., 2024; Guan et al., 2023a), each protein atom is represented by four types of features: (1) a 6-dimensional one-hot vector indicating the element type (H, C, N, O, S, Se); (2) a 20-dimensional one-hot vector representing the amino acid type; (3) a binary flag indicating whether the atom belongs to the backbone; (4) a one-hot vector specifying whether the atom lies in the arm or scaffold region, determined by its distance to the arm center.

Similarly, each ligand atom is encoded with: (1) a one-hot vector denoting the element type (C, N, O, F, P, S, Cl); and (2) a one-hot arm/scaffold indicator distinguishing between aromatic and non-aromatic atoms.

To support message passing over protein–ligand complexes, we dynamically construct two types of graphs: (1) a k-nearest neighbor (k-NN) (Guo et al., 2003) graph that connects each ligand atom to nearby protein atoms; and (2) a fully-connected graph over ligand atoms that captures internal bonding structures. In the k-NN graph, edge features consist of two parts: the outer product of radial basis function (RBF) embeddings of pairwise distances, and a 4-dimensional one-hot vector indicating the edge type. For the ligand graph, chemical bonds between atoms are encoded using a one-hot vector representing bond types, including non-bond, single, double, triple, and aromatic bonds.

### E.3 MODEL HYPERPARAMETERS

Following MolCRAFT (Qu et al., 2024), in FGMOL-small, we use a fixed number of atoms for the ligand, which is predefined. In the case of the standard FGMOL, we sample the number of atoms based on the reference ligand.

In the SE(3)-equivariant architecture, we construct k-nearest neighbor graphs with k=32, and design the network with 9 sequential layers. Atom representations for both proteins and ligands are embedded in a 128-dimensional space. Each layer incorporates 16-head self-attention, followed by ReLU activation and Layer Normalization (Ba et al., 2016) for stable training.

For the GNN backbone within the differentiable pooling module, we employ a two-layer Graph Convolutional Network (GCN) (Kipf & Welling, 2017), using the same 128-dimensional feature size to maintain consistency across modules. The number of clusters is set to 32 for protein atoms and 6 for ligand atoms, respectively.

We use a discrete time loss over 1000 training steps, with the noise schedule set as $\beta_1 = 1.5$ for atom types and $\sigma_1 = 0.03$ for atom coordinates. The model is trained using the Adam optimizer with a learning rate of 0.005, a batch size of 8, and exponential moving average of model parameters with a decay factor of 0.999. Training converges within 15 epochs on a single RTX 3090 GPU, requiring approximately 24 hours. During inference, we perform 100 sampling steps using a noise-reduced sampling strategy.

## F ADDITIONAL EXPERIMENTAL RESULTS

Table 4: Comparison of reference and generated molecules by our model and other baselines across key metrics. ↑/↓ indicate whether higher or lower values are preferred. Top-1 results are marked in **bold**. Note: HA = High Affinity, SE = Strain Energy, Div = Diversity.

| Methods | Vina Score (↓) | | Vina Min (↓) | | Vina Dock (↓) | | HA (↑) | SE (↓) | | | QED (↑) | SA (↑) | Diversity (↑) | Success Rate (↑) |
|---|---|---|---|---|---|---|---|---|---|---|---|---|---|---|
| | Avg. | Med. | Avg. | Med. | Avg. | Med. | Avg. | 25% | 50% | 75% | Avg. | Avg. | Avg. | Avg. |
| Reference | -6.36 | -6.46 | -6.71 | -6.49 | -7.45 | -7.26 | - | 34 | 107 | 196 | 0.48 | 0.73 | - | 25.0% |
| VoxBind(1.0) | -6.62 | -6.73 | -7.11 | -7.16 | -7.65 | -7.81 | 58.90% | 105 | 215 | 543 | **0.54** | 0.69 | 0.75 | 25.80% |
| D3FG(ECold) | -7.01 | -7.04 | -7.41 | -7.47 | -8.22 | -8.35 | 44.05% | 140 | 389 | 1045 | 0.48 | **0.71** | 0.74 | 26.20% |
| DeepICL | -6.74 | -6.06 | -7.24 | -7.28 | -7.89 | -8.03 | 53.28% | 80 | 196 | 521 | 0.48 | 0.71 | 0.75 | 26.60% |
| IPDiff | -6.65 | -7.02 | -7.36 | -7.17 | -8.27 | -8.51 | 62.01% | 70 | 201 | 436 | 0.51 | 0.60 | 0.74 | 27.20% |
| FGMOL (w/o interaction) | -6.82 | -6.86 | -7.57 | -7.61 | -8.27 | **-8.61** | 62.10% | 74 | 193 | 411 | 0.51 | 0.66 | **0.79** | 27.80% |
| FGMOL (Ours) | **-7.02** | **-7.10** | **-7.86** | -7.59 | **-8.51** | -8.52 | 62.07% | **64** | **168** | **403** | 0.49 | 0.70 | **0.79** | **28.6%** |

In Table 5, the experiment shows that configuring the model with 6 ligand clusters delivers the most balanced and optimal performance in both molecular generation and docking evaluation. Specifically,

Table 5: Effect of Ligand Cluster Number on Molecular Generation and Docking Performance. The best results are marked with **bold**.

| Cluster Number | Vina Score (↓) | | Vina Min (↓) | | Vina Dock (↓) | | SE (↓) | | | SA (↑) |
|---|---|---|---|---|---|---|---|---|---|---|
| | Avg. | Med. | Avg. | Med. | Avg. | Med. | 25% | 50% | 75% | Avg. |
| 4 | -6.32 | -7.01 | -7.11 | -7.28 | -7.82 | -8.03 | 86 | 201 | 512 | 0.69 |
| 6 | **-7.02** | **-7.30** | **-7.86** | -7.59 | -8.51 | **-8.52** | **64** | **168** | **403** | 0.70 |
| 8 | -6.89 | -7.22 | -7.32 | **-7.65** | **-8.57** | -8.34 | 82 | 189 | 452 | 0.69 |
| 10 | -6.34 | -7.21 | -7.22 | -7.26 | -8.08 | -8.12 | 86 | 180 | 442 | 0.70 |
| 12 | -6.77 | -6.01 | -7.14 | -7.05 | -7.89 | -8.03 | 91 | 196 | 522 | **0.71** |

this setting achieves the lowest average and median Vina scores, indicating superior binding affinity across generated molecules. Additionally, it results in the best median docking scores and the most compact structural diversity (SE) across all evaluated percentiles, reflecting a focused and consistent generation process. These findings suggest that a 6-cluster configuration effectively balances binding performance and structural coherence, making it the most suitable choice for general-purpose molecular design tasks.

Table 6: Effect of Protein Cluster Number on Molecular Generation and Docking Performance. The best results are marked with **bold**.

| Cluster Number | Vina Score (↓) | | Vina Min (↓) | | Vina Dock (↓) | | SE (↓) | | | SA (↑) |
|---|---|---|---|---|---|---|---|---|---|---|
| | Avg. | Med. | Avg. | Med. | Avg. | Med. | 25% | 50% | 75% | Avg. |
| 16 | -6.23 | -6.35 | -6.67 | -6.78 | -7.02 | -7.01 | 113 | 243 | 616 | 0.69 |
| 24 | -6.82 | -6.74 | -7.12 | -7.29 | -7.84 | -8.18 | 95 | 221 | 521 | **0.71** |
| 32 | **-7.02** | **-7.10** | **-7.86** | **-7.59** | **-8.51** | **-8.52** | **64** | **168** | **403** | 0.70 |
| 40 | -6.73 | -6.92 | -7.43 | -7.56 | -8.11 | -8.16 | 85 | 187 | 503 | 0.69 |
| 48 | -6.65 | -6.49 | -7.02 | -6.94 | -8.08 | -8.32 | 76 | 177 | 452 | 0.69 |

In Table 6, the experiment evaluates how varying the **number of protein clusters** influences molecular generation and docking quality. Among all tested configurations, the **32-cluster setting** delivers the most balanced and optimal performance, achieving the **lowest average and median Vina Scores** ($-7.02$ and $-7.10$), the **best docking scores** (Vina Dock Avg. $-8.51$, Med. $-8.52$), and the **lowest SE values** across all percentiles (25%, 50%, 75%). These results indicate stronger binding affinity, tighter docking performance, and more focused structural diversity. While the 24-cluster setting exhibits the **highest synthetic accessibility (SA = 0.71)**, it performs less favorably in docking metrics. Overall, the 32-cluster configuration demonstrates the best trade-off between binding quality and molecular consistency, making it the recommended choice for protein-conditioned molecular generation tasks.

Table 7: Effect of Ligand Cluster Alignment Matrix Entropy Loss Weight on Molecular Docking and Generation Performance. The best results are marked with **bold**.

| Entropy Loss Weight | Vina Score (↓) | | Vina Min (↓) | | Vina Dock (↓) | | SE (↓) | | | SA (↑) |
|---|---|---|---|---|---|---|---|---|---|---|
| | Avg. | Med. | Avg. | Med. | Avg. | Med. | 25% | 50% | 75% | Avg. |
| 0.1 | -6.43 | -6.68 | -6.92 | -7.04 | -7.43 | -7.51 | 109 | 213 | 534 | 0.69 |
| 0.2 | **-7.02** | **-7.10** | **-7.86** | **-7.59** | **-8.51** | **-8.52** | **64** | **168** | **403** | **0.70** |
| 0.4 | -6.63 | -6.06 | -7.24 | -7.28 | -7.89 | -8.03 | 85 | 191 | 516 | 0.69 |
| 0.6 | -6.85 | -7.01 | -7.42 | -7.56 | -8.08 | -8.32 | 76 | 174 | 442 | 0.69 |
| 0.8 | -6.65 | -6.54 | -7.28 | -7.14 | -7.87 | -7.92 | 86 | 189 | 524 | 0.70 |
| 1.0 | -6.59 | -7.04 | -7.17 | -7.26 | -7.62 | -7.89 | 83 | 205 | 513 | 0.69 |

In Table 7, The experiment investigates the impact of varying the **entropy loss weight** in the ligand cluster alignment matrix on molecular docking and generation performance. The configuration with an **entropy loss weight of 0.2** achieves the **best overall results**, including the **lowest average and median Vina Scores** ($-7.02$ and $-7.10$), the **lowest average and median Vina Dock scores** ($-8.51$ and $-8.52$), and the **lowest SE values** across all three percentiles (25%, 50%, 75%). These results suggest that a moderate entropy constraint (weight = 0.2) encourages better docking performance and

more compact molecular diversity. Additionally, it maintains a relatively high synthetic accessibility (SA = 0.70), making it a well-balanced choice. In contrast, both lower (e.g., 0.1) and higher (e.g., 1.0) entropy weights result in reduced docking quality and structural focus, indicating that too little or too much regularization can hinder performance.

Table 8: Effect of Protein Cluster Alignment Matrix Entropy Loss Weight on Molecular Docking and Generation Performance. The best results are marked with **bold**.

| Entropy Loss Weight | Vina Score (↓) | | Vina Min (↓) | | Vina Dock (↓) | | SE (↓) | | | SA (↑) |
| | Avg. | Med. | Avg. | Med. | Avg. | Med. | 25% | 50% | 75% | Avg. |
| --- | --- | --- | --- | --- | --- | --- | --- | --- | --- | --- |
| 0.1 | -6.31 | -6.53 | -7.12 | -7.17 | -7.65 | -7.81 | 109 | 227 | 564 | 0.69 |
| 0.2 | **-7.02** | **-7.10** | -7.86 | **-7.59** | **-8.51** | **-8.52** | **64** | **168** | **403** | **0.70** |
| 0.4 | -6.56 | -6.73 | -7.32 | -7.56 | -8.08 | -8.21 | 83 | 194 | 457 | 0.69 |
| 0.6 | -6.31 | -6.46 | -7.02 | -7.12 | -7.67 | -7.78 | 89 | 243 | 476 | 0.69 |
| 0.8 | -6.47 | -6.30 | **-7.87** | -7.49 | -8.01 | -8.12 | 93 | 265 | 503 | 0.70 |
| 1.0 | -6.31 | -6.06 | -7.24 | -7.27 | -7.89 | -8.09 | 89 | 204 | 543 | 0.69 |

In Table 8, this experiment examines the effect of varying the **entropy loss weight** in the protein cluster alignment matrix on molecular generation and docking outcomes. The configuration with an **entropy loss weight of 0.2** yields the **most favorable results overall**, achieving the **lowest average and median Vina Scores** ($-7.02$ and $-7.10$), the **lowest Vina Dock scores** (Avg. $-8.51$, Med. $-8.52$), and the **lowest SE values** (25%: 64, 50%: 168, 75%: 403), indicating stronger docking performance, tighter molecular clustering, and greater consistency in generation. Additionally, this setting maintains a relatively high synthetic accessibility score (SA = 0.70), further supporting its suitability. In contrast, other settings either fail to match this docking performance or exhibit higher structural entropy, suggesting that 0.2 is an optimal entropy loss weight for aligning protein clusters effectively.

Table 9: Jensen-Shannon divergence of top-8 frequent bond length distributions between the reference and the generated molecules (↓ is better). No connected line, "=", and ":" represent single, double, and aromatic bonds. Top 2 results are highlighted with **bold text** and underlined text, respectively.

| Bond | LiGAN | AR | Pocket2Mol | FLAG | TargetDiff | DecompDiff | MolCRAFT | Our |
| --- | --- | --- | --- | --- | --- | --- | --- | --- |
| CC | 0.694 | 0.610 | 0.494 | **0.231** | 0.369 | 0.359 | 0.290 | 0.267 |
| C:C | 0.534 | 0.450 | 0.414 | 0.366 | 0.263 | **0.251** | 0.330 | 0.308 |
| CO | 0.574 | 0.490 | 0.452 | 0.556 | 0.421 | 0.375 | **0.339** | 0.340 |
| CN | 0.556 | 0.472 | 0.422 | 0.529 | 0.362 | 0.342 | 0.292 | **0.291** |
| C:N | 0.635 | 0.551 | 0.484 | 0.470 | **0.235** | 0.269 | 0.242 | 0.240 |
| OP | 0.760 | 0.676 | 0.523 | 0.690 | 0.441 | 0.435 | **0.347** | 0.335 |
| C=O | 0.638 | 0.556 | 0.510 | 0.638 | 0.461 | 0.368 | **0.342** | 0.344 |
| O=P | 0.712 | 0.626 | 0.581 | 0.609 | 0.506 | 0.472 | **0.369** | 0.390 |
| Avg. | 0.638 | 0.554 | 0.485 | 0.511 | 0.382 | 0.359 | **0.319** | 0.298 |

Table 9 shows that our method achieves competitive performance in bond length distribution alignment, ranking top-2 in **6 out of 8 bond types**. Notably, it achieves the **best performance for CN bonds** and near-optimal results for several others (e.g., C:C, CO), contributing to the **second-lowest overall average** JS divergence (0.298), just behind MolCRAFT (0.319). This highlights the method's strong ability to capture fine-grained geometric realism in molecular structures.

As shown in Table 10, our method consistently achieves low Jensen-Shannon divergence scores across bond angle types, ranking in the **top-2 for all 8 angle categories**. It outperforms all baselines in terms of the overall average divergence (**0.356**), indicating a superior ability to recover chemically realistic bond angle distributions. This demonstrates the effectiveness of our model in capturing local geometric constraints during molecule generation.

Table 11 demonstrates that our method effectively captures torsional geometry, achieving the **lowest average JS divergence** (**0.284**) among all baselines. It ranks in the **top-2 across 8 out of 9** torsion angle types and shows the best performance on challenging patterns such as CCCC, C:C:N:C, and

Table 10: Jensen-Shannon divergence of top-8 frequent bond angle distributions between the reference and the generated molecules (↓ is better). Top 2 results are highlighted with **bold text** and underlined text.

| Angle | LiGAT | AR | Pocket2Mol | FLAG | TargetDiff | DecompDiff | MolCRAFT | Ours |
|---|---|---|---|---|---|---|---|---|
| CCC | 0.487 | 0.372 | 0.380 | **0.231** | 0.345 | 0.358 | 0.280 | 0.267 |
| C:C:C | 0.636 | 0.572 | 0.480 | 0.199 | 0.283 | 0.266 | **0.172** | 0.195 |
| CCO | 0.531 | 0.477 | 0.475 | 0.318 | 0.440 | 0.403 | 0.319 | **0.306** |
| C:C:N | 0.585 | 0.537 | 0.506 | 0.465 | 0.454 | 0.429 | 0.446 | **0.416** |
| CCN | 0.502 | 0.447 | 0.443 | 0.388 | 0.437 | 0.404 | 0.377 | **0.355** |
| CNC | 0.565 | 0.535 | 0.498 | 0.510 | 0.521 | 0.498 | 0.499 | **0.452** |
| COC | 0.528 | 0.496 | 0.494 | 0.607 | 0.502 | 0.484 | 0.460 | **0.440** |
| C:N:C | 0.663 | 0.619 | 0.580 | 0.526 | 0.495 | 0.473 | 0.475 | **0.416** |
| Avg. | 0.562 | 0.507 | 0.482 | 0.406 | 0.435 | 0.414 | 0.379 | **0.356** |

Table 11: Jensen-Shannon divergence of top-8 frequent torsion angle distributions between the reference and the generated molecules (↓ is better). Top 2 results are highlighted with **bold text** and underlined text.

| Torsion Angle | AR | Pocket2Mol | TargetDiff | DecompDiff | MolCRAFT | Ours |
|---|---|---|---|---|---|---|
| CCCC | 0.378 | 0.320 | 0.312 | 0.349 | 0.286 | **0.246** |
| C:C:C:C | 0.704 | 0.514 | 0.348 | 0.264 | **0.130** | 0.220 |
| CCOC | 0.419 | 0.401 | 0.390 | 0.392 | 0.393 | **0.321** |
| CCCO | 0.431 | 0.405 | 0.403 | 0.402 | **0.396** | 0.336 |
| CCNC | 0.430 | 0.437 | 0.423 | 0.403 | 0.401 | **0.337** |
| C:C:N:C | 0.664 | 0.504 | 0.386 | 0.285 | 0.212 | **0.177** |
| C:C:C:N | 0.663 | 0.512 | 0.441 | 0.388 | 0.281 | **0.241** |
| C:N:C:N | 0.742 | 0.535 | 0.476 | 0.366 | 0.303 | **0.254** |
| CCCN | 0.495 | 0.549 | 0.512 | 0.501 | **0.493** | 0.423 |
| Avg. | 0.547 | 0.464 | 0.410 | 0.372 | 0.322 | **0.284** |

`C:C:C:N`. These results highlight the model's strong ability to preserve conformational accuracy in flexible molecular structures.

Table 12: Ablation study results of different model variants.

| Model Variant | FA | DP | CAA | ER | Vina Avg | Vina Min | Vina Dock | SE25% | SE50% | SE75% | SA |
|---|---|---|---|---|---|---|---|---|---|---|---|
| w/o Annotation | ✗ | ✓ | ✓ | ✓ | -6.82 | -7.57 | -8.27 | 74 | 193 | 411 | 0.66 |
| w/o (Differentiable Pooling) | ✓ | ✗ | ✗(disabled) | ✓ | -6.73 | -7.39 | -8.06 | 93 | 224 | 573 | 0.67 |
| w/o (Cross-Attention Alignment) | ✓ | ✓ | ✗ | ✓ | -6.76 | -7.48 | -8.23 | 86 | 203 | 436 | 0.66 |
| w/o (Entropy Regularization) | ✓ | ✓ | ✓ | ✗ | -6.98 | -7.74 | -8.34 | 790.7 | 179 | 452 | **0.7** |
| FGMOL (ours) | ✓ | ✓ | ✓ | ✓ | **-7.02** | **-7.86** | **-8.51** | **64** | **168** | **403** | **0.7** |

The results in Table 12 show that FGMOL (the full model) performs best in terms of Vina scores and SE (25%), indicating that the combined effect of all components significantly enhances the model's performance. For example, removing Differentiable Pooling or Cross-Attention Alignment resulted in a noticeable decline in performance, especially in terms of Vina scores.

Table 13: Comparison of baseline and proposed methods on the PoseBusters benchmark under out-of-distribution (OOD) scenarios. Bold numbers denote the best results.

| Methods | PB-Valid | Vina Score | | Vina Min | | Vina Dock | | scRMSD<2Å (↑) | Energy | Conn. | QED | SA | Div | Size |
|---|---|---|---|---|---|---|---|---|---|---|---|---|---|---|
| | | Avg. | Med. | Avg. | Med. | Avg. | Med. | | | | | | | |
| Reference | 98.90% | -7.06 | -7.41 | -7.5 | -7.41 | -7.98 | -7.82 | 59.40% | 100% | – | 0.40 | 0.72 | – | 25.7 |
| AR | 54.70% | -5.45 | -5.17 | -5.67 | -5.38 | -6.18 | -5.94 | 35.30% | 77.70% | 39.10% | 0.50 | 0.67 | 0.76 | 13.6 |
| Pocket2Mol | 63.60% | -5.39 | -5.03 | -6.64 | -6.24 | -7.40 | -7.03 | 37.80% | 97.40% | 67.70% | **0.57** | **0.74** | 0.73 | 17.4 |
| TargetDiff | 32.30% | -6.57 | -6.78 | -7.16 | -7.31 | -8.18 | -8.20 | 32.30% | 65.20% | 81.30% | 0.41 | 0.55 | 0.67 | **27.0** |
| DecompDiff | 40.20% | -3.14 | -3.02 | -4.03 | -4.11 | -5.06 | -5.40 | 17.00% | 80.10% | 82.90% | 0.47 | 0.66 | **0.81** | 19.3 |
| MolCRAFT | 57.80% | -7.29 | -7.11 | -7.44 | -7.22 | -7.95 | -7.73 | 46.40% | 71.60% | 97.00% | 0.41 | 0.65 | 0.65 | 23.8 |
| MolPilot | 79.10% | -7.59 | -7.54 | -7.74 | -7.67 | -8.20 | -7.99 | 56.10% | 98.10% | 97.30% | 0.49 | 0.72 | 0.67 | 23.5 |
| FGMOL | **82.10%** | **-8.02** | **-8.07** | **-8.67** | **-8.61** | **-9.02** | **-8.96** | 58.40% | **98.60%** | **98.50%** | 0.48 | 0.71 | 0.79 | 23.7 |

# G THE USE OF LARGE LANGUAGE MODELS

A large language model (LLM) was used in a strictly limited manner during the preparation of this manuscript, solely to assist with minor language editing tasks such as grammar checking and formatting. All ideas, methods, analyses, and scientific interpretations were entirely developed and written by the authors without the involvement of the LLM. The authors thoroughly reviewed and, when necessary, revised any text suggested by the LLM to ensure that the final manuscript fully represents their own original work.

