# OpenReview forum: "Beyond Geometry: Functionally Grounded Molecule Generation for Structure-Based Drug Design"
_ICLR.cc/2026/Conference — Submitted to ICLR 2026_

### Official Review · Reviewer_NMEf · 2025-10-29

**Soundness:** 3
**Presentation:** 3
**Contribution:** 2
**Rating:** 4
**Confidence:** 3

**Summary:**

This paper introduces FGMOL (Functionally Grounded Molecule Generation Network) for structure-based drug design (SBDD), arguing that current generative models overly focus on geometric fit while neglecting crucial biochemical functional interactions (e.g., hydrogen bonds, hydrophobic interactions, π-π stacking) between proteins and ligands. FGMOL aims to address this by integrating functional interactions into the generation process via a unified structure-function alignment framework. The method proposes three key components: 1) Interaction-Aware Embedding, which explicitly annotates protein atoms with interaction types and feeds them into SE(3)-equivariant networks; 2) Interaction-Informed Motif Alignment, using differentiable clustering and Sinkhorn matching to align protein-ligand functional motifs ; and 3) Interaction-Guided Generation using a Bayesian Flow Network (BFN), which models coordinates and atom types in continuous space conditioned on the alignment via cross-attention. Experiments on the CrossDocked2020 benchmark show that FGMOL improves binding affinity, significantly reduces ligand strain energy (>20%) compared to SOTA methods, and maintains high synthetic accessibility.

**Strengths:**

- Explicitly encoding interaction types into protein atom features and processing them with SE(3)-equivariant networks provides a principled way to make the model aware of functional sites .

- The Interaction-Informed Motif Alignment step, using differentiable pooling to identify pharmacophore-like motifs and Sinkhorn normalization for alignment, introduces a valuable inductive bias for functional complementarity .

- Leveraging BFN for continuous generation guided by functional priors via cross-attention is a technically sound approach to promote physically plausible and functionally relevant structures .

**Weaknesses:**

- The paper seems to rely solely on atomwise features for added functional interactions. However, critical interactions, such as hydrogen bonds, are inherently pairwise features. This representation might be insufficient. Furthermore, the source of the 'function annotations' is unclear. Are these annotations extracted from the ground-truth ligand, or are they user-specified? If they are derived from the ground-truth ligand, this raises a significant concern about potential information leakage (or data leakage), where the model might be unfairly exposed to target information.
- The proposed model (FGMOL) incorporates several complex components, such as cross-attention and alignment mechanisms, and also utilizes all-atom modeling for the protein. Intuitively, these additions should introduce significant computational overhead. However, the paper claims that its sampling efficiency is merely 'similar' to MOLCRAFT. This finding is counterintuitive. Can the authors provide a more detailed breakdown or explanation for why these sophisticated components do not lead to a noticeable decrease in sampling efficiency compared to the MOLCRAFT baseline?

**Questions:**

- Could the authors elaborate on the source and method used for generating the functional interaction annotations? How sensitive is the model to the quality of these annotations and which functions contribute most?

---

### Official Review · Reviewer_NxuY · 2025-10-31

**Soundness:** 2
**Presentation:** 3
**Contribution:** 3
**Rating:** 4
**Confidence:** 3

**Summary:**

The authors present FGMol, a de novo ligand generation method. FGMol uses interaction-aware features and training to guide the generation process, yielding better results than traditional AI SBDD methods.

**Strengths:**

Paper is written well and easy to understand. Although adding additional features such as HBD, HBA, etc is not new, the BFN and matching of the ligand clusters and protein clusters is a good choice to model interactions between protein subpockets and ligand motifs.

**Weaknesses:**

While FGMol has the best docking scores and druglikeness metrics, it would be nice to see error bars. For example, D3FG appears to do only marginally worse than FGMol on Vina metrics (table 4). In addition, the SA and QED seem to be on par with other methods. Overall, while I appreciate the thoughtful design choices of the method, I’m not convinced of their efficacy compared to baselines.

**Questions:**

Could the authors elaborate on how the learned molecular motifs are different from existing work? Specifically, I believe DecompDiff does something similar (they call them ‘scaffolds’ in their paper), inferring motifs of a molecule and letting that guide the learning process.

Why are the results split into two tables? Specifically, tables 1 and 4. they appear to be showing the same metrics, why not combine them into 1 table? In addition, it looks like the baseline methods in the appendix (D3FG and IPDiff) are better than the baseline methods in the main text. IPDiff and D3FG are approaching the performance of FGMol; error bars showing statistical significance of results would be appreciated here.

---

### Official Review · Reviewer_BkqB · 2025-10-31

**Soundness:** 2
**Presentation:** 3
**Contribution:** 2
**Rating:** 2
**Confidence:** 4

**Summary:**

This submission explores the problem of structure-based drug design adopting a generative molecule model approach. The authors investigate two While most existing methods model only the geometry of the target protein and the ligand, the authors propose a method called FGMOL which also models the functional interactions between the two molecules. Domain knowledge about potential functions of atoms are integrated as priors into a Bayesian Flow Network (BFN). Experiments on the commonly used CrossDocked dataset were performed to compare FGML with several baselines, demonstrating gains in particular a higher proportion of high-affinity molecules.

**Strengths:**

Consideration of functional interactions promises to generate stronger and more stable interactions between protein and ligand.

Sampling in a continuous parameter space avoids the need of switching between continuous and discrete aspects of molecule representations.

Experiments were performed on the commonly used CrossDocked benchmark dataset.

**Weaknesses:**

Several related works have already explored interaction-guided approaches for structure-based drug design, including IPDiff (Huang et al., 2024b) and FLOWR (Cremer et al., 2025). The authors do not adequately differentiate their method from these existing methods.

Several aspects of the design of FGMOL are questionable:
-  It seems to me that interaction-aware features should be considered not only for the protein but also for the ligand.
- You state that you are also clustering ligand atoms based on similar interaction semantics, but that is not possible if you do not have interaction-aware features for ligands.
- Binding motifs could consist of interactions of multiple types, but FGMOL restricts clusters to groups of atoms with similar interaction semantics.

FGMOL lacks technical novelty since it îs largely a combination of existing methods, I.e. BFNs, SE(3)-equivariant NNs, the clustering method of Yang et al., 2018, the sampling approach of Qu et al., 2024.

The authors claim their cross-attention mechanism promotes "synthetically feasible" structures, but provide no evidence for this. The SA scores in Table 1 show only marginal improvements (0.70 vs 0.69 for MolCRAFT). They should benchmark using computational retrosynthesis tools like AiZynthFinder to actually demonstrate synthesizability. See Gao et al. (2024) "Reframing structure-based drug design model evaluation via metrics correlated to practical needs" for relevant discussion.

It is well known that docking scores and interaction counts correlate with molecular size - larger molecules naturally form more contacts and get better scores. The authors report improved Vina scores and interaction counts (Figure 3b-d, Figure 4) but do not control for molecule size. Without normalizing for molecule size, we do not know whether the docking score improvements are real or just due to generating larger molecules.

**Questions:**

1) What are the commonalities and the differences between FGMOL and Diff and FLOWR?

2) Do you really use interaction-aware features only for the protein? If so, why, and how can you then cluster ligand atoms with similar interaction semantics?

3) What is the ligand efficiency (LE = -Docking Score / HAC, where HAC = heavy atom count) of the compared methods?  See Hopkins et al. (2004) "Ligand efficiency: a useful metric for lead selection" Drug Discovery Today 9(10):430-431 and Kuntz et al. (1999) "The maximal affinity of ligands" PNAS 96(18):9997-10002.

---

### Official Review · Reviewer_2Zp9 · 2025-11-01

**Soundness:** 3
**Presentation:** 2
**Contribution:** 2
**Rating:** 4
**Confidence:** 4

**Summary:**

This paper presents a diffusion-based framework for learning functional protein–ligand representations that move beyond purely geometric modeling. The authors propose a unified latent diffusion model that conditions ligand generation on protein context, aiming to capture functional binding preferences rather than static spatial configurations. The work is conceptually relevant to the community’s shift from geometry-only to function-aware modeling. The integration of diffusion processes with binding-site features is promising and shows potential for future extensions in structure-based drug design.

Overall, the idea is novel and valuable, but the paper’s current form lacks comparative breadth, clarity of presentation, and some necessary experimental validation to convincingly support its claims.

**Strengths:**

1.	Timely problem formulation. The paper clearly recognizes the limitations of geometry-only paradigms (e.g., EquiBind, DiffDock) and attempts to incorporate more functional information. This is a meaningful and forward-looking direction for generative modeling in drug discovery.
2.	Methodological potential. The BFN-based conditioning strategy is technically sound, and the formulation seems general enough to extend to binding affinity prediction or multi-ligand environments.
3.	Qualitative promise: The generated ligand examples and the functional conditioning strategy suggest that the model may indeed be learning context-aware protein–ligand distributions, which could be impactful if verified on standard benchmarks.
4.	Conceptual clarity: The conceptual introduction and motivation are clearly stated, and the paper generally reads well from a methodological standpoint.

**Weaknesses:**

- Missing Comparisons with Relevant Baselines

While the paper claims to move “beyond geometry,” it omits comparison with several geometic-aware ligand generative methods that have already been evaluated in CBGBench or similar frameworks. DiffSBDD, DiffBP, VoxelBind, and GraphBP should be considered as references for assessing context-conditioned ligand generation. Without such baselines, it is unclear whether the proposed method outperforms or even matches these diffusion-based models on the same functional benchmarks. I strongly recommend evaluating on CBGBench, with more metrics against DiffSBDD/DiffBP.

- Weak Figure and Visual Presentation

Figure 2 is very difficult to interpret. The text is tiny, the schematic elements are disproportionately spaced, and large blank areas make it visually unbalanced. Readers cannot easily infer the data flow, diffusion/BFN steps, or conditioning mechanism from this figure. Given that this figure is central to the method, poor readability significantly undermines the perceived maturity and polish of the paper. I recommend redrawing it with consistent color coding for protein vs. ligand vs. latent variables, readable font (>8 pt in print), clear annotation of the process.

- Ambiguity in Feature Construction and Generation Consistency

The description of interaction embeddings is unclear. You mention that interaction types are represented as one-hot encodings and embedded as input features, but in most docking/complex datasets, such embeddings are computed by external software (e.g., PLIP or PyRosetta) — which requires both the protein and the ligand structures as input.
That raises several questions:
	 1) During generation, the ligand does not yet exist — so what interaction embedding is used as input to the model?
	 2) If the embedding is a learned latent variable, how is it initialized or constrained?
	3) Is there a post-generation validation step that checks whether the predicted interactions correspond to physically meaningful patterns (H-bonds, π–π stacking, salt bridges, etc.)?

In previous work such as CBGBench, generated structures were re-scored using molecular interaction software to verify whether the distribution of recovered interactions matched experimental distributions. This paper does not include such an analysis, making it difficult to claim that the model indeed preserves functional interaction types or site-specific contact statistics. Or providing more visualization on certain diseases' targets to show that the generated molecules keep the preferred functional interactions of protein pockets.

**Questions:**

See weakness

---

### Author Response · Authors · 2025-12-01
**Response to all reviewers and ACs**

We sincerely thank the reviewers for their valuable and constructive comments. Their suggestions, including the request for more benchmarks and additional evaluation metrics, have helped us further improve the quality of this work. Below, we provide our consolidated responses to the raised questions.

**Q1: On the concern that interactions are inherently pairwise**

We agree that many key protein–ligand interactions, such as hydrogen bonds, are intrinsically pairwise. Our method explicitly leverages this property.
Analogous to ligand design driven by the 3D shape of a protein pocket—where the compatibility is also fundamentally pairwise—we use interaction information to guide ligand generation. When provided with the protein pocket’s interaction patterns, our model learns to generate ligands that satisfy the required pairwise interaction relationships.
Thus, our representation is conceptually aligned with the pairwise nature of biochemical interactions.

**Q2: On the concern regarding the source of functional annotations and potential data leakage**

To prevent information leakage, we clearly separate interaction extraction and utilization into distinct stages:

* Training phase: We obtain interaction annotations on the protein pocket side only using PLI predictions from known protein–ligand complexes.

* Generation phase: In realistic use cases, ligand-specific interaction details are typically unavailable.
Therefore, we define a standardized set of interaction categories and assign interaction conditions solely based on protein atoms, without relying on any ground-truth ligand information.

This design ensures that our generation process remains free of target-specific leakage and is applicable in practical scenarios where no native ligand is known.

**Q3: On the differences from DiffDecompt, IPDiff, and FLOWR**

DiffDecompt, IPDiff, and FLOWR fundamentally differ from our method:

* DiffDecompt: Models only geometric complementarity and does not explicitly represent interaction types.

* IPDiff: Uses an implicit potential field that cannot capture fine-grained interaction-type distinctions, and its performance is highly sensitive to the ligand-number prior (as shown in our Appendix).

* FLOWR: Uses interaction masks derived directly from ground-truth PLI/PLIF matrices and performs an inpainting-style reconstruction, rather than de novo interaction-driven ligand generation.

In contrast, our method explicitly models protein-side interaction types and enables de novo ligand generation guided by desired interaction patterns, independent of any native ligand.
This yields a fundamentally different formulation and capability compared with prior works.

---

We sincerely thank all reviewers and the ACs for their time. Their insights have greatly strengthened our work.
We will incorporate the comments and release a revised version of the manuscript, and we look forward to sharing the improved version with the community.

---

### Meta-Review · Area_Chair_Navn · 2026-01-07

**Summary:**

Looking at the reviews, several consistent themes emerge across the four reviewers' assessments. The most fundamental concern centers on how the method handles interaction annotations and whether there's potential data leakage. Multiple reviewers point out that key interactions like hydrogen bonds are inherently pairwise phenomena, yet the proposed approach relies on atomwise features. This raises questions about whether the representation adequately captures the physics of protein-ligand binding. The uncertainty about where these functional annotations come from during generation is particularly problematic since the ligand doesn't yet exist when you're trying to generate it.

The lack of comprehensive benchmarking appears repeatedly in the reviews. Reviewer 2Zp9 specifically notes the absence of comparisons with established geometry-aware methods like DiffSBDD, DiffBP, VoxelBind, and GraphBP, and suggests evaluation on CBGBench. This reviewer also raises concerns about whether the improvements might simply reflect generating larger molecules rather than genuinely better interactions, since docking scores naturally correlate with molecular size. The point about needing ligand efficiency metrics rather than raw docking scores is well-taken.

The differentiation from existing interaction-aware methods remains unclear. Both reviewers question how FGMOL differs meaningfully from IPDiff, FLOWR, and DiffDecompt. The authors' response attempts to distinguish these but the reviewers seem unconvinced that the technical contributions are sufficiently novel given that the method largely combines existing components like BFNs, SE(3)-equivariant networks, and established clustering approaches.

Statistical rigor is lacking throughout the evaluation. Reviewer NxuY specifically requests error bars since the performance differences with baselines like D3FG appear marginal, particularly for synthetic accessibility and druglikeness metrics. Without statistical significance testing, it's difficult to assess whether the reported improvements are meaningful or within noise.
The presentation quality needs improvement, particularly Figure 2 which Reviewer 2Zp9 describes as difficult to interpret with tiny text and poor visual balance. More broadly, there's confusion about why results are split between Table 1 and Table 4 when they appear to show similar metrics.

Finally, several reviewers note that while the paper claims to generate synthetically feasible structures, the evidence is weak. The synthetic accessibility scores show only marginal improvements, and there's no validation using actual retrosynthesis tools to demonstrate whether these molecules could realistically be made in a laboratory.

**Reviewer Concerns:**

No point-by-point rebuttal was provided to the reviewers' concerns. A general rebuttal was provided by the authors. The general rebuttal addressed the most direct technical concerns about representation and data leakage reasonably well, but sidestepped the methodological critiques about evaluation rigor, size bias controls, and comprehensive benchmarking that would actually determine whether the method represents a genuine advance over existing approaches.

**Reviewer Scores:**

Reviewer 2Zp9 - Initial Score: 4
This reviewer would likely maintain their score of 4 or possibly drop to 2.  Their primary concern about missing comparisons with relevant baselines like DiffSBDD, DiffBP, and evaluation on CBGBench receives no acknowledgment in the rebuttal. The figure quality issue they raised extensively also goes unaddressed.

Reviewer BkqB - Initial Score: 2
This reviewer would almost certainly maintain their score of 2. Their most critical concern about molecular size bias and the need for ligand efficiency metrics goes completely unaddressed. This is a fundamental methodological flaw in the evaluation that the rebuttal doesn't even acknowledge.

Reviewer NxuY - Initial Score: 4
This reviewer might stay at 4. Their main concerns were about statistical significance and error bars, plus clarification on how motifs differ from DecompDiff. The rebuttal doesn't address the error bars request at all, which remains a significant weakness.

Reviewer NMEf - Initial Score: 4
This reviewer would likely move up slightly to 6. Their two main concerns about information leakage and computational overhead receive direct responses. The Q2 response about separating training and generation phases to prevent leakage directly addresses their primary worry about data contamination. While the computational overhead question doesn't get answered in the rebuttal, this seemed like a secondary curiosity rather than a deal-breaker concern.

---

### Decision · Program_Chairs · 2026-01-26

Reject